



# Relevance of acoustic methods to quantify bedload transport and bedform dynamics in a large sandy-gravel bed river

Jules Le Guern[1], Stéphane Rodrigues[1,2], Thomas Geay[3], Sébastien Zanker[4], Alexandre Hauet[4], Pablo Tassi[5,6], Nicolas Claude[5,8], Philippe Jugé[7], Antoine Duperray[1], Louis Vervynck[1].

[1]UMR CNRS CITERES, University of Tours, France.
[2]Graduate School of Engineering Polytech Tours, University of Tours, France.
[3]BURGEAP R&D, Grenoble, France.
[4]EDF, Division Technique Générale, Grenoble, France.
[5]EDF R&D – National Laboratory for Hydraulics and Environment (LNHE), Chatou, France.
[6]Saint-Venant Laboratory for Hydraulics, Chatou, France.
[7]CETU Elmis Ingénieries, University of Tours, Chinon, France.
[8]EDF, Centre Ingénierie Hydraulique, La Motte Servolex, France.

*Correspondence to*: Jules Le Guern (leguern@univ-tours.fr).

**Abstract**

**Despite the inherent difficulties to quantify its value, bedload transport is essential to understand fluvial systems. In this study, we assessed different indirect bedload measurement techniques with a reference direct bedload measurement in a section of a large sandy-gravel bed river. Acoustic Doppler Current Profiler (aDcp), Dune Tracking Method (DTM) and hydrophone measurement techniques were used to determine bedload transport rates using calibration with the reference method or using empirical formula. Results show that the hydrophone is the most efficient and accurate method to determine bedload flux in the Loire River. Even though parameters controlling self-generated noise of sediments still need to better understood, the calibration determined in this study allows a good approximation of bedload transport rates. Moreover, aDcp and hydrophone measurement techniques are both able to continuously measure bedload transport associated to bedform migration.**

## 1. Introduction

Worldwide, rivers are in crisis (Vorösmarty et al., 2010). While changes in flow characteristics and fragmentation are well known (Grill et al., 2019), the impacts of human activities on the sediment budgets is yet underrepresented (Kondolf et al., 2018). The quantification of bedload transport is a key element to understand, manage and restore the physical and ecological functioning of fluvial systems. It constitutes a prerequisite to the accurate estimation of global sediment budgets delivered by rivers to oceans (Syvitski and Milliman, 2007), to better understand bedform dynamics in river channels (Best, 1988; Bertoldi et al., 2009; Rodrigues et al., 2015; Claude et al., 2014) and to





reproduce satisfactorily morphodynamic processes with numerical modelling (Mendoza et al. 2017; Cordier et al.,

33  2020).

However, in large rivers, this parameter remains difficult to estimate mainly due to the human and material
resources required to correctly quantify its measurement. Among the available tools, indirect measurement
techniques are promising alternatives (Gray et al., 2010) to direct measurements that are often cumbersome to
implement, and can be time-consuming and perilous. Since the 2000s, numerous studies were proposed to process
the signal captured by acoustic Doppler current profilers (aDcp) as a tool for determining the apparent bedload
velocity (Rennie et al., 2002; Rennie and Villard, 2004; Rennie and Millar, 2004; Kostaschuk et al., 2005; Villard et
al., 2005; Gaeuman and Jacobson 2006; 2007; Holmes et al., 2010; Ramooz and Rennie, 2010; Latosinski et al.,
2017). The use of passive acoustic instruments has also been widely used to quantify bedload transport. These
techniques have been developed through the application of measurement tools such as geophones or
hydrophones, but their domain of applicability is restricted to the study of rivers with coarse-sizes sediments (Barton
et al., 2010; Hilldale et al., 2014; Marineau et al., 2016; Geay et al., 2017).
In sandy-gravel bed rivers, the presence of bedforms is generally used to indirectly estimate bedload transport
(Simons et al., 1965). Single beam (Peters, 1978; Engel and Lau, 1980) or multibeam echosounders (Nittrouer et
al., 2008; Leary and Buscombe, 2020) are standard techniques used to determine morphological parameters such
as bedform height, wavelength and celerity. Moreover, these acoustic measurements are carried out
simultaneously with sediment sampler measurements (Claude et al., 2012) to calibrate the signal with a direct
reference although the latter are intrusive and characterized by a low spatial representativeness. These drawbacks
can therefore limit the applicability of these measurement techniques, in particular for large lowland rivers.
In this work, we compare the efficiency of active (aDcp, echosounder) and passive (hydrophone) acoustic
techniques is assessed for the quantification of bedload transport in a reach of the Loire River (France), which is
characterized by the presence of migrating bars and superimposed dunes (Le Guern et al., 2019b).
The main objectives of this study were: 1) to compare indirect measurements techniques with a direct measurement
technique commonly employed in large sandy-gravel bed rivers (isokinetic samplers) at determining bedload
sediment transport rates; 2) to estimate the capacity of acoustic signals to detect the bedload axes on relatively
wide cross-sections for various discharge conditions; and 3) to investigate the capabilities of hydrophones and
aDcps for capturing bedload transport on bedforms.

## 2. Study site

The study site is located near Saint-Mathurin-sur-Loire, in the lower reach of the Loire River (France), approximately
150 km upstream of the mouth of the Loire River. The study reach is 2.5 km long, 500 m wide, nearly straight, with
a bed slope of 0.02 % (Fig. 1). The riverbed is composed of a mixture of siliceous sands and gravels with a median



diameter of 0.9 mm. The width-to-depth ratio ranges from 120 to 550 depending on discharge variations. The mean
annual discharge at the Saumur gauging station (approx. 30 km upstream) is 680 m³.s⁻¹, with a 2-years flood return
period equal to 2700 m³.s⁻¹. Surveys were conducted during various hydrological conditions, with flow discharges
ranging from 200 to 2400 m³.s⁻¹ (Fig. 2a).
Bars are characterized by an average wavelength of 1300 m, corresponding to approximately three times the
channel width. The mean bar height is 1.5 m. At submerged conditions, bars can migrate with a celerity of 0.5 to 2
meters per day. During floods, the bar celerity can increase up to 4 meters per day (Le Guern et al., 2019a). During
floods, dunes are superimposed to bars, whose height, wavelength and mean celerity are approximately of 0.3 m,
4.4 m and 32 meters per day, respectively.

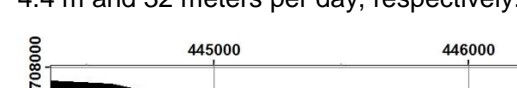

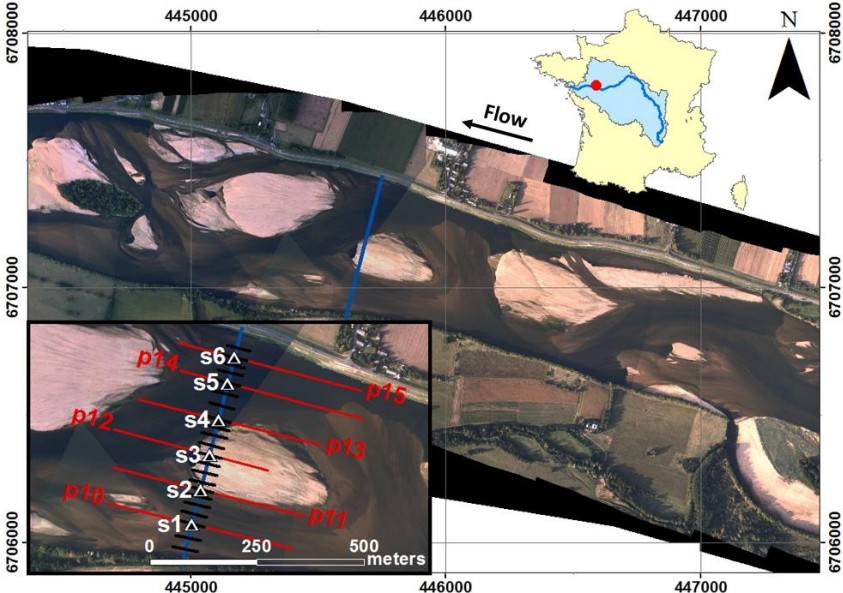


**Fig. 1: Aerial view of the study site in 2017 (courtesy of Dimitri Lague, University of Rennes, France) with location of**
**sampling points (white triangles) on the sediment transport gauging cross section (blue line), bathymetric profiles (red**
**lines) and theoretical hydrophone drifts (black lines).**
**3. Materials and methods**
Direct measurements of bedload sediment transport rates were performed using isokinetic samplers (BTMA, see
below) adopted here as reference. This classical approach was retained to evaluate three indirect acoustic
methods: the apparent bedload velocity assessed from aDcp measurements, the dune tracking method (DTM)
inferred using single-beam echosounding, and the self-generated noise (SGN) of sediments measured using a
hydrophone. A total of 72 surveys were performed from October 2016 to May 2020 (discharge ranging between
210 m³.s⁻¹ and 2290 m³.s⁻¹) including 43 surveys with isokinetic samplers presented on Fig. 2a.





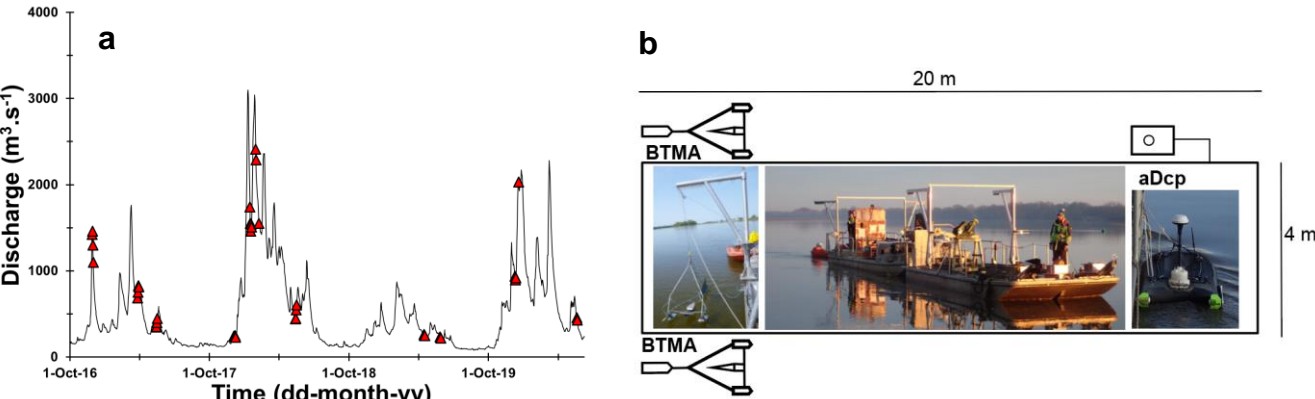

**Fig. 2: (a), distribution of surveys along the hydrograph of Saumur gauging station located about 30 km upstream the study site. (b), Scheme of the main boat and disposition of monitoring facilities.**

**3.1. Bedload rates obtained using isokinetic samplers**

Bedload transport rates were measured using two synchronized isokinetic Bedload Transport Meter Arnheim (BTMA) samplers mounted on a 20 meter-long boat stabilized using two anchors (Fig. 2b). These two samplers were deployed on 6 sampling points (S1 to S6) distributed along a cross section (Fig. 1). On each sampling point, collected BTMA values were integrated over at least 2 minutes. The unit bedload rate for each sampling point was obtained by averaging the volumes of sediment collected using Eq. (1):

$$q_s BTMA = \frac{V}{b} \, \alpha \, \varepsilon \, \rho_s \times 10^3; \qquad\qquad (1)$$

where $q_s BTMA$ is the unit bedload transport rate (g.s⁻¹.m⁻¹), $\alpha$ is the trap efficiency factor based on calibration ($\alpha$=2), $V$ is the mean volume of the sediment catch (m³.s⁻¹), $b$ is the width of the mouth of the instrument ($b$=0.085 m), $\rho_s$ is the volumetric mass of sediment (2650 kg.m⁻³) and $\varepsilon$ is the sediment concentration (assumed to be 0.65). Suggested values of $\alpha$ and $b$ were adopted from Boiten (2003). Sampler positions and sampling quality were controlled by using two cameras mounted on the BTMAs. Sediment samples were analysed using the standard sieving technique (Folk and Ward, 1957) to determine the grain size distribution (GSD) using the tool "GRADISTAT" developed by Blott and Pye (2001).

**3.2. Apparent bedload velocity from aDcp**

Simultaneously with the BTMA measurements, two aDcps were positioned on the boat (Fig. 2b). Measurements were performed using a Sontek Riversurveyor M9 (bi-frequency, 1 and 3 MHz) and/or a Teledyne RD Instruments Rio Grande (1.2 MHz). Sampling time ranged between 5 and 190 minutes. ADcp were coupled with a RTK GPS Magellan ProFlex 500 receiving position corrections via the Teria network (centimeter level accuracy). The aDcp measurement allow the use of both empirical approach and calibration approach for comparison with sediment sampler measurements. The apparent bedload velocity $V_a$ was estimated from the bottom tracking signal that allow



the identification and the position of the river bed. In case of mobile bed, the Doppler shift of the backscattered
acoustic pulse of the bottom track depends on to the boat velocity and to the bed velocity. According to Rennie et
al. (2002), the apparent bedload velocity can be estimated using:
$V_a = V_{GPS} - V_{BT}$;                                                                                                     (2)
where $V_{GPS}$ and $V_{BT}$ are respectively the boat velocity according to GPS reference and bottom track. When the
GPS signal was poor or missing, the apparent velocity was considered directly equal to the boat velocity according
to bottom track reference because measurements were performed in a static position (representing 15% of the
dataset). Following Jamieson et al. (2011), the apparent velocity was calculated for the North and East velocity
components (respectively $V_{aE}$ and $V_{aN}$), giving better results especially in areas where inconsistent directions and
low magnitudes of bedload velocity were found: $V_a = \sqrt{V_{aE}^2 + V_{aN}^2}$.
To avoid compass and GPS issues, and to eliminate the effect of residual lateral displacement of the anchored
boat, the apparent bedload velocity was projected onto the flow direction using:
$V_{a\,proj} = V_a \cdot cos\left(\frac{w_{dir\,GPS} - b_{dir\,BT}}{180} \cdot \pi\right)$;                                  (3)
with $w_{dir\,GPS}$ the water direction with GPS reference and $b_{dir\,BT}$ the boat direction with the bottom track reference (in
degree). Eq. (3) gives a value of apparent bedload transport velocity for each time step (about 1 s) that was
averaged to obtain a value for each sampling point. According to Rennie et al. (2002), the bedload transport rate
per unit width ($q_s\,ADCP$, g.s$^{-1}$.m$^{-1}$) can be computed from two different models:
$q_s ADCP = \frac{4}{3} \rho_s\, r\, V_a \times 10^3$;                                                                         (4)
$q_s ADCP = V_a d_s (1-\lambda)\rho_s$;                                                                                         (5)
In Eq. (4), $r = D_{50}/2$ is the particle radius and the active layer thickness ($d_s$) is considered as a constant, with $D_{50}$ is
the median sediment diameter (m), $\rho_s$ is the sediment density (2650 kg.m$^{-3}$). In Eq. (5), $\lambda$ is the porosity of the
active transport layer considered as a constant and equal to 0.35, and the van Rijn (1984) formulation was adopted
to compute the active layer thickness as a function of the hydraulic condition and sediment grain size:
$d_s = \frac{0.3\, D_*^{0.7}\, T^{0.5}}{D_{50}}$;                                                                                (6)
$T = \frac{\left(u'_*\right)^2 - (u_{*cr})^2}{(u_{*cr})^2}$;                                                                      (7)
$u'_* = \frac{\bar{u}}{5.75\, log\left(\frac{12d}{3D_{90}}\right)}$;                                                            (8)
where $T$ is the transport stage parameter that reflects the sediment mobility, $u'_*$ is the bed shear velocity related to
the grain (m.s$^{-1}$), $d$ is the mean water depth (m), $D_{90}$ is the 90$^{th}$ percentile of the sediment grain size (m), , $\bar{u}$ is the
mean water velocity measured from aDcp (m.s$^{-1}$) and $u_{*cr}$ is the critical bed shear velocity (m.s$^{-1}$) calculated from
the Shields curve (Van Rijn, 1984) and function of grain size through the scaled particle parameter $D_*$:



$D_* = D_{50} \left[ \frac{(s-1)g}{v^2} \right]^{\frac{1}{3}};$                                                                    (9)
where $g$ is the acceleration of the gravity (m.s⁻²), $v$ the kinematic viscosity (m².s⁻¹) and $s$ the sediment density ratio.
For the range of grain size of this study, $u_{*cr}$ is computed as follows:
$10 < D_* \leq 20; \; u_{*cr} = \left[ 0.04 \, D_*^{-0.1} (s-1) g D_{50} \right]^{0.5};$                                      (10)
$20 < D_* \leq 150; \; u_{*cr} = \left[ 0.013 \, D_*^{0.29} (s-1) g D_{50} \right]^{0.5};$                                   (11)
To assess the capability of the aDcp to detect bedforms through the evolution of apparent bedload velocity, 3
surveys were conducted by positioning the aDcp 0.6 m above the river bed. This experimental scheme was adopted
to avoid lateral movements of the boat, to be as close as possible to the river bed, and to reduce the space between
beams. This configuration permitted to fix the insonified surface for each beam to about 0.0046 m² and a distance
of 0.56 m between opposed beams, and could allow a better understanding of the apparent bedload velocity
gradient along bedforms. These surveys were performed during several hours (from 2.1 h to 4.7 h) to see more
than one dune lee side pass under the device. The value of apparent bedload velocity was smoothed by using a
moving windows with an average of 500 points (approximately 500 seconds) to remove the noise from the raw
dataset. In the present study, all negative values were excluded from the comparison with BTMA measurements
(16% of apparent velocity values).
**3.3. Bathymetrical echosounding and dune tracking method**
A single beam echosounder Tritech PA500 (0.5 kHz) coupled with a RTK GPS LEICA Viva GS25 were used for
high-frequency bathymetric surveys to determine bar and dune morphodynamics along 6 longitudinal profiles
(about 400 m long) centred on sampling points indicated in Fig. 1. Dune height ($H_D$) and wavelength ($\lambda_D$) were
estimated using the Bedform Tracking Tool (BTT) based on the zero-crossing method (Van der Mark and Blom,
2007). Dune celerity ($C_D$) was estimated with the Dune Tracking Method (DTM, Simons et al., 1965; Engel and
Lau, 1980) following the dune crests between two subsequent bathymetric surveys for a mean interval time equal
to 40 minutes. The interval time needs to be adjusted with discharge because of the dune celerity variation from
one survey to another. The determination of a proxy to evaluate sediment transport directly from DTM
measurements is difficult. A semi-empirical equation was used to compare bedload transport rates with the
reference measurement. The computed dune parameters were used to calculate the unit bedload transport rate
($q_s DTM$, g.s⁻¹.m⁻¹) using the formula by Simons et al. (1965):
$q_s DTM = (1-\lambda) \, \rho_s \, H_D \, C_D \, \beta \times 10^3;$                                                         (12)
where $H_D$ is the mean dune height along the profile (m), $C_D$ is the median dune celerity (m.s⁻¹) and $\beta$ is the bedload
discharge coefficient equal to 0.5 for a perfect triangular dune shape. The $\beta$ coefficient neglects the volume of
bypassing material from previous dunes or exchanges between bedload and suspended load (Wilbers, 2004). Due



to its large variability (Van den Berg, 1987; Ten Brinke et al., 1999; Wilbers, 2004), the sensibility of the bedload
transport rate was assessed for $\beta$=[0.33; 0.57], as proposed in the literature (Engel and Lau, 1980; Wilbers, 2004).
Considering the accuracy of the bathymetrical echosounding and the representativeness of dune celerity, only
profiles with a mean dune height of 0.1 m and more than 10 dunes are considered.
**3.4. Hydrophone and acoustic power**
Passive acoustic monitoring was performed with a Teledyne RESON Hydrophone TC4014-5 (sensitivity of -180
dB) plugged into an EA-SDA14 card from RTSYS Company. This device has a large frequency range from 0.015
to 480 kHz, with a linear response until 250 kHz (±3dB). The hydrophone has been deployed following the protocol
proposed by Geay et al. (2020). A total of 22 longitudinal profiles was defined on the sediment transport gauging
section (see Fig. 1). The boat was positioned directly upstream the sediment transport gauging section and left
adrift at flow velocity. Depending on the water depth, the hydrophone was installed at a constant depth between
0.4 and 0.7 m below the water surface. Data acquisition was stopped after the boat crossed the sediment transport
gauging section, so drift duration ranged between 15 to 140 seconds, depending on the flow velocity (mean time
of 31 s). The acoustic power ($P$) for each drift was computed by integrating the median Power Spectral Density
($PSD$) over a range of frequency comprised between $f_{min}$ (15 kHz) and $f_{max}$ (350 kHz) (Geay et al., 2020):
$P = \int_{f_{min}}^{f_{max}} PSD(f)df$ ;                                                 (13)
The minimum frequency was chosen to avoid hydrodynamic and engine noises, while the maximum frequency was
set by the upper limit frequency of the device and was adjusted related to $PSD$. Finally, the nearest hydrophone
drift for each BTMA sampling point was selected. Several tests were carried out to ensure that these acoustic
power variations were not related to the distance between the hydrophone and the river bed. As no theoretical
formula has been developed to estimate bedload rates from hydrophone measurements, only the calibration
approach was implemented.
**4. Results**
**4.1. Comparison between acoustics and direct bedload transport rate measurements**
The BTMA dataset is composed of 135 unit bedload rates calculated from 2628 individual sediment sampling. That
represents an average of 19 samples on each sampling point to compute unit bedload rates (minimum of 5 and
maximum of 57 samples). Bedload rates measured using the BTMAs ranged between 0.01 and 268 g.s$^{-1}$.m$^{-1}$. The
standard deviation of unit bedload rates increased with discharge with a mean value of 33 g.s$^{-1}$.m$^{-1}$. This illustrates
the spatio-temporal variability of sediment transport induced by bedform migration.





The aDcp dataset is composed of 98 simultaneous measurements of apparent bedload velocity and BTMA
samplings (Fig. 3a). The mean apparent bedload velocity is 0.022 m.s$^{-1}$ and the maximum value was 0.11 m.s$^{-1}$. A
Reduced Major Axis (RMA) regression has been computed between these two variables with a coefficient of
determination (COD) R² equal to 0.48:
$q_s = 3545\ V_a^{1.25}$;                                                                                    (14)
As shown in Fig. 3a, this site-specific calibration of the Loire River is in good agreement with the Rees, Missouri
and Fraser rivers (Rennie et al., 2017). It also suggests that the experimental relationship between $V_a$ and $q_sBTMA$
is similar for similar material (in terms of grain size).

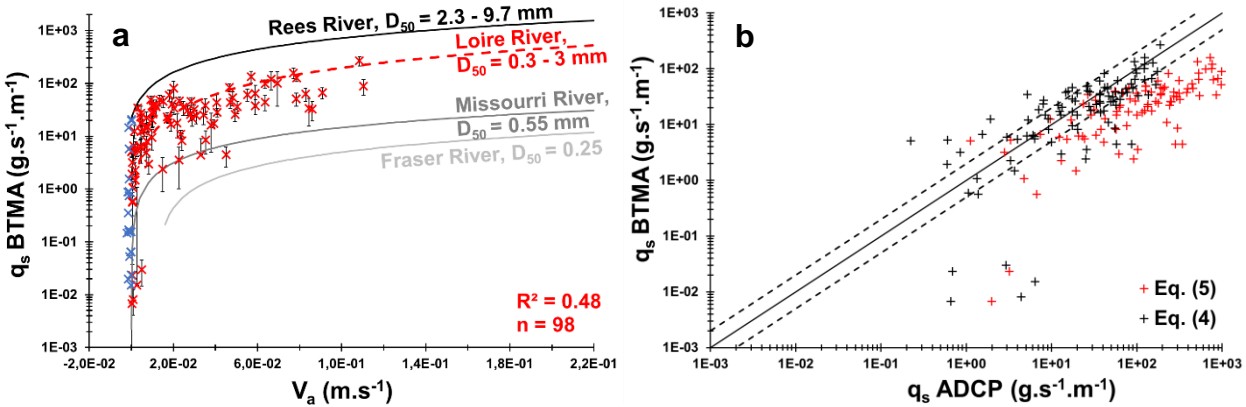


**Fig. 3: (a), unit bedload transport rates measured with BTMA samplers as a function of the apparent bedload velocity**
**measured with aDcp. Red dashed line represents the RMA regression of the Loire River. Comparison with other site-**
**specific calibration curves (Rennie et al. 2017). Blue marks represent negatives apparent bedload velocity excluded**
**from this regression. (b), log/log correlation between bedload rates measured with BTMA sampler and calculated using**
**Eq. (4) and Eq. (5). Solid black line represents the perfect correlation and dashed black lines represents a factor 2 of**
**the perfect correlation.**
Bedload transport rates were calculated considering the concentration and the thickness of active layer as constant.
In order to evaluate the accuracy of a method against a reference, the discrepancy ratio is classically employed in
the literature (Van Rijn, 1984; Van den Berg, 1987; Batalla, 1997). This ratio is defined as the ratio between the
bedload rate estimated with the indirect method and the bedload rate using BTMA. Approximately 57% of the
computed bedload transport rate (Eq. 4) is within the discrepancy ratio (Figure 3b), while only 14% when using the
Van Rijn definition of the active layer thickness (Eq. 5). According to these results, considering the active layer
thickness proportional to the median sediment grain size seems to be a good approximation to determine bedload
rate, especially for bedload rate greater than 1 g.s$^{-1}$.
It appears difficult to estimate bedload rates only from dune celerity by making a direct relation between dune
celerity and bedload transport rates measured with BTMA. Estimation of bedload transport rates from dune
morphology has been performed by using empirical formula of Simons et al. (1965) (Eq. 12). The dataset is
composed of 49 DTM profiles with associated BTMA samples. The mean dune height and length vary from 0.1 to





0.5 m, and 1.3 to 12 m, respectively. The median dune celerity varies between 13 and 61 m.d⁻¹. According to Fig.
4, bedload rates estimated with a discharge coefficient $\beta$ of 0.33 are in agreement with BTMA bedload rates with
67% of values in a factor 2, whereas 49% for a discharge coefficient of 0.57. The Engel and Lau, (1980) definition
of the discharge coefficient is better adapted for dune shape of the Loire River which are characterized by mean
steepness ($H_D/L_D$) of 0.05 (in line with other observations on the Loire River, Claude et al., 2012; Rodrigues et al.,
2015; Wintenberger et al., 2015).

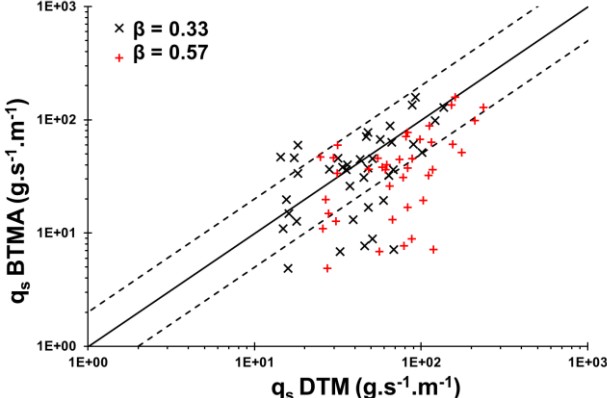


**Fig. 4: log/log correlation between bedload rates measured with BTMA samplers and bedload rates calculated using**
**Eq. (12). Solid black line represents the perfect correlation and dashed black lines represents a factor 2 of the perfect**
**correlation.**
Even if the statistical representativeness is lower than other methods (n=37), the RMA regression between the
acoustic power and BTMA sampling is better (R²=0.70) and 60% of values varying between a factor 2 (Fig. 5a). A
new equation for estimate sediment transport from acoustic power is proposed:
$P = 6.6 \times 10^{10} \, q_s^{1.32};$          (15)
This calibration curve is similar to observations performed by Geay et al. (2020) on 14 study sites distributed on 11
different rivers despite the use of different instruments (sampler and hydrophone) and the integration of median
PSD over a wider range of frequency in the present study.
The acoustic power corresponding to the integration of the spectrum over a range of frequency is related to grain
size (Thorne, 1985) and sediment kinematics (Gimbert et al., 2019). To analyze the effect of sediment mobility on
the acoustic power, the transport stage parameter (Van Rijn, 1984) is calculated. The power law adjusted between
these two parameters evidences a positive evolution of the acoustic power with sediment mobility (Fig. 5b).





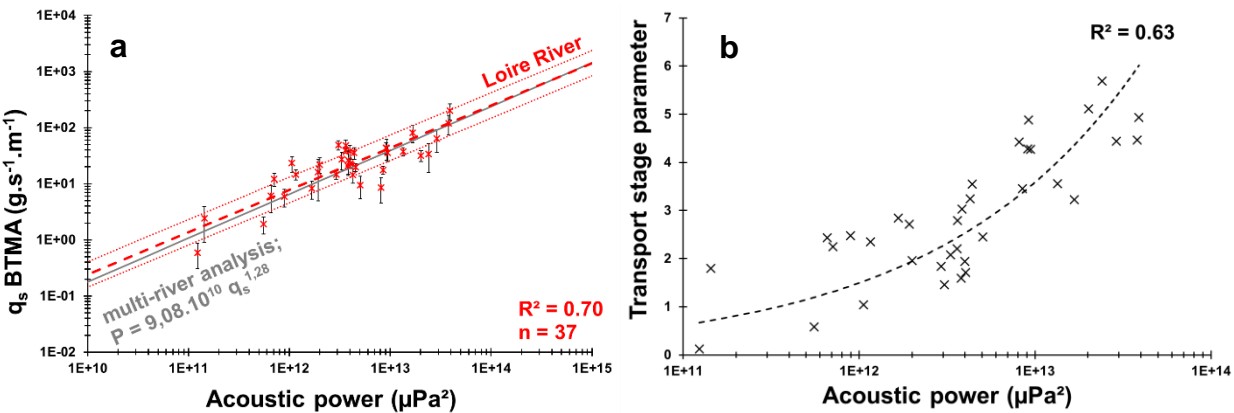

**Fig. 5: (a), unit bedload rates measured with BTMA samplers as a function of acoustic power measured with hydrophone. Dashed red lines represents the RMA regression with envelopes curves of a factor 2 of the bedload rates. Comparison with Geay et al. (2020). (b), transport stage parameter (from Van Rijn, 1984) as a function of acoustic power.**

The comparison can be made between indirect methods to discuss the acceptability of the BTMA reference. The apparent bedload velocity and the acoustic power are not well-correlated with mean dune morphological parameters (dune celerity and dune height). The aDcp method is measuring the apparent velocity of the grain being transported from the stoss to the lee side of a dune. It must be noted that apparent bedload velocity is higher than dune celerity with about a factor 100, whereas the grain size ($D_{50}$) is smaller than dune height with the same order. Therefore, sediments that are 100 times smaller than dune height allows the dune migration with a celerity 100 times smaller than their own celerity. On the other hand, the apparent bedload velocity is positively correlated with the acoustic power. The RMA regression model explains 76% of the dataset dispersion (Fig. 6a).

Before focusing on the spatial distribution of unit bedload rates, total bedload rates are calculated by interpolating unit bedload rates between sampling points on the cross section for each method. The RMA regression established between BTMA bedload rates and water discharge explain 71% of the dataset dispersion (Fig. 6b) with 77% of the values varying in a factor 2. The dispersion of bedload rates is higher for low water discharge (under mean annual discharge, 800 $m^3.s^{-1}$). Bedload rates are estimated from Eq. (12), Eq. (14) and Eq. (15), for the DTM, the aDcp and the hydrophone, respectively. Both the hydrophone and DTM bedload rates are less scattered with 96% of values in the discrepancy ratio, whereas 73% for the aDcp.



Earth **Surface**
**Dynamics**
Discussions



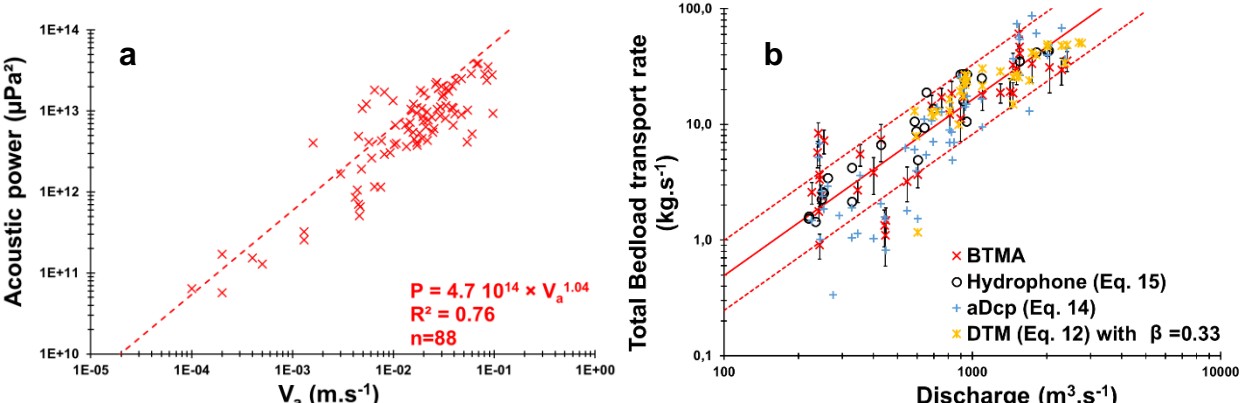

266

**Fig. 6: (a), acoustic power as a function of apparent bedload velocity. (b), Cross section integrated bedload transport rates as a function of discharge.**

**4.2. Spatial distribution of bedload in a sandy gravel-bed river with migrating bedforms**

**4.2.1. Determination of bedload transport axes on a cross section using acoustics methods**

To compare the spatio-temporal distribution of bedload transport rates, sediment transport gauging was performed on the same cross section for all surveys and for various discharge conditions. Two surveys with contrasting discharge conditions and different bed configurations are presented to illustrate the capacity of acoustics methods to determine bedload axes in a river reach characterized by the presence of macroform and superimposed mesoforms (*sensu lato,* Jackson, 1975).

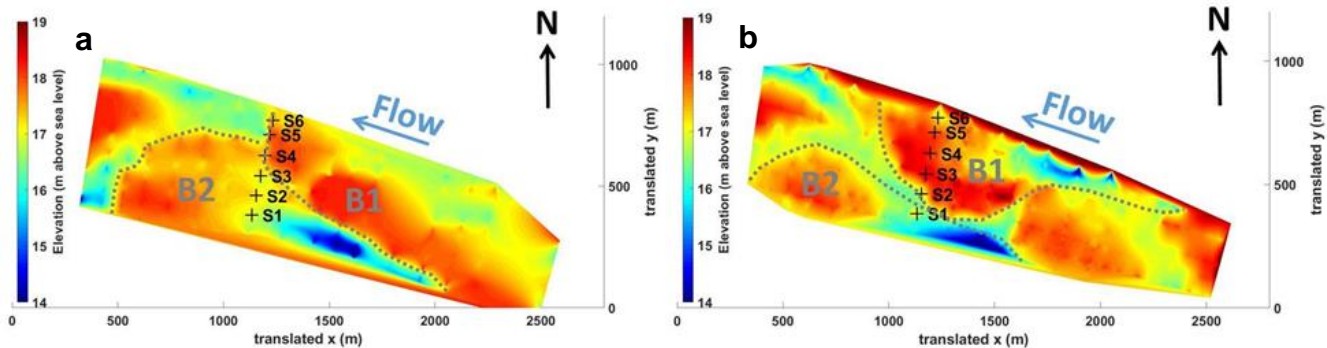

276

**Fig. 7: Bathymetric Digital Elevation Models (obtained using natural neighbours interpolation) showing location of sampling points with respect to bars location during: (a), survey of the 17/05/2018 (Q=604 $m^3.s^{-1}$) and (b), survey of the 19/12/2019 (Q=2050 $m^3.s^{-1}$).**

In May 2018, a bar (B1. Fig. 7a) was located just upstream the sediment gauging section from the center to the right part of the channel. In the left part of the channel, BTMA sampling was done on the stoss side of another bar (B2, Fig. 7a). Consequently, bedload rates were gradually rising from the center of the channel (2 $g.s^{-1}.m^{-1}$, S4) to the left part of the channel (15 $g.s^{-1}.m^{-1}$, S1) except for the DTM (Fig. 8a). The intensity of bedload transport rates





was evaluated for each acoustic signal from regression equations established above (Eq. 12, Eq. 14 and Eq. 15,
for DTM, aDcp and hydrophone, respectively). ADcp and hydrophone signals followed the same evolution as the
BTMA measurement. In the right part of the channel, there was no reference measurements (S5 and S6) but all
acoustic signals followed the same trend (increasing bedload transport rates). The bedload rates estimated with
the DTM were lower than the reference in the left part of the channel. This can be explained by the reduced number
of dunes in this area that caused a higher uncertainty in dune celerity determination. In the right part, the proximity
of the bar front induced lower bedload transport rates measured with aDcp and hydrophone. DTM integrates
sediment dynamics over a longitudinal profile that does not necessarily reflect the bedload transport conditions at
local scale. Due to the lee effect provided by the proximity of the bar front, dunes were not present downstream of
the bar and only dunes located on the stoss side of the bar were used to calculate the mean dune celerity. ADcp
underestimates whereas hydrophone overestimates BTMA measurements.

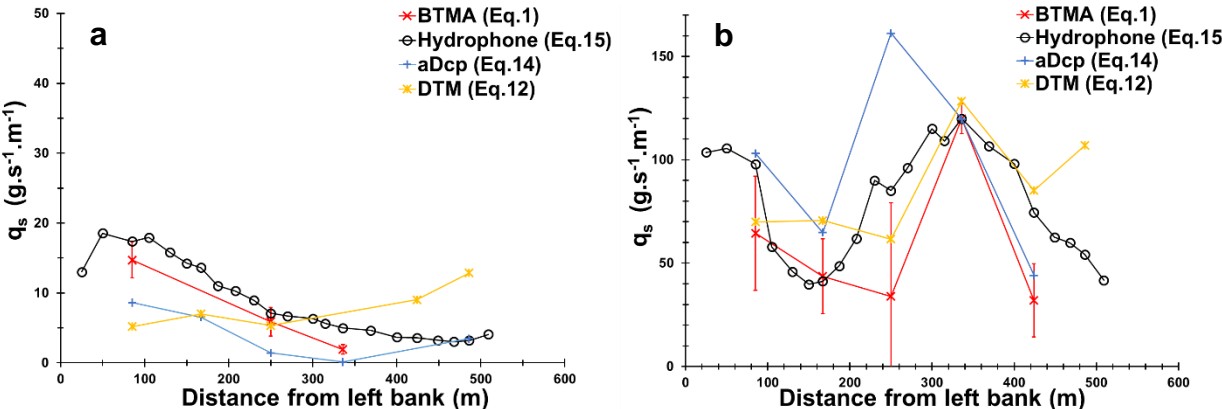


**Fig. 8: Lateral distribution of unit bedload rates assessed from different methods for two surveys performed: (a), the**
**17/05/2018 (Q=604 m³.s⁻¹) and (b), the 19/12/2019 (Q=2050 m³.s⁻¹), respectively.**
In December 2019 (Fig. 8b), discharge was higher (2050 m³.s⁻¹) and measured bedload rates ranged between 32
and 120 g.s⁻¹.m⁻¹. Due to the bar migration, the bed configuration was different. Bar B1 migrated on the sediment
gauging cross section. As a consequence, sampling points S3 to S6 were located on the stoss side of bar B1 (Fig.
7b). The sampling point S2 was located just downstream the bar front where velocity and sediment transport rates
were lower (Fig. 7b). The high spatial resolution of the hydrophone measurements confirmed that the preferential
bedload axis was located between 250 and 450 m from the left bank (Fig. 8b). For this survey, acoustic signals
(i.e. acoustic power, apparent bedload velocity) followed the same evolution pattern as isokinetic samplers along
the cross section except for S3. Bedload transport rates determined with the DTM did not follow the trend of bedload
rates determined with aDcp and hydrophone at the proximity of bar front and near the bank as in the previous
survey (S2 and S6). The hydrophone model overestimated the sediment transport in comparison with the BTMAs
for S1, S3 and S5.

Earth **Surface**
**Dynamics**
Discussions

### 4.2.2. Sediment transport processes on bedforms analyzed from aDcp and hydrophone

The aDcp computed bedload rates evolved according to bedform location for fixed measurements performed on dunes of height ranging between about 0.05 m and 0.2 m (Fig. 9a and 9b). Higher bedload rates values were found on the crest of the dune and lower values in the trough. The amplitude of bedload rates between crest and trough for low flow conditions (Fig. 9a) ranged between 42 $g.s^{-1}.m^{-1}$ and 69 $g.s^{-1}.m^{-1}$. For higher flow conditions, it varied between 43 $g.s^{-1}.m^{-1}$ and 111 $g.s^{-1}.m^{-1}$ (Fig. 9b). The aDcp power regression (Eq. 14) did not allow the calculation of bedload transport rates due to negative apparent bedload velocity. This is the case downstream the lee face of dunes (Fig. 9a, between 19-42 min., 104-107 min., 185-193 min., and 227-230 min.; Fig. 9b, between 48-55 min. and 153-162 min.). The mean time recorded between two successive dune crests was 1 hour.

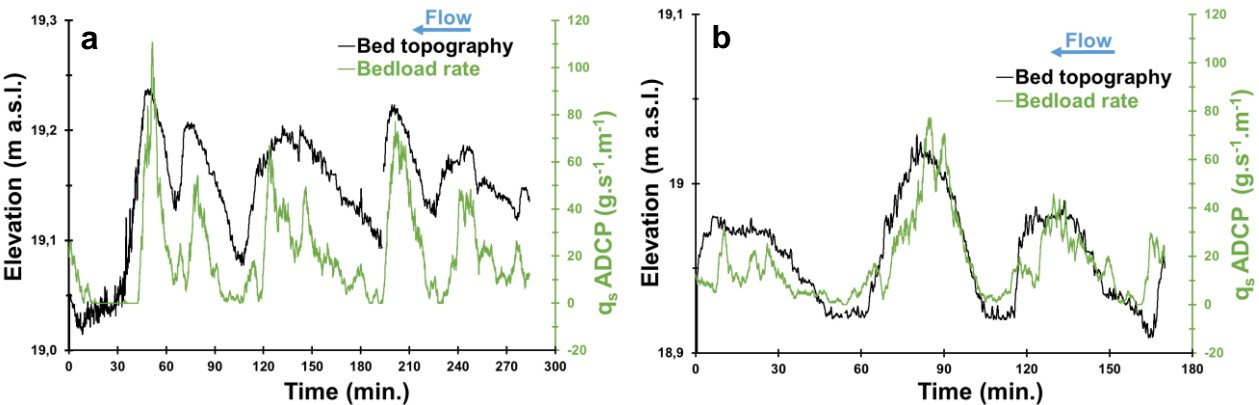

**Fig. 9: Bedload rates calculated using Eq. (14) and bed topography obtained during a static measurement performed using an aDcp. (a), survey done the 20/05/2020 (Q=470 $m^3.s^{-1}$; mean water depth = 1.04 m) and (b), survey done the 29/05/2019 (Q=210 $m^3.s^{-1}$; mean water depth = 0.85 m).**

Hydrophone drifts showed that the longitudinal evolution of acoustic power can be correlated with changes in elevation of the riverbed due to dune and bar presence. For instance, in the presence of a 2 meter high bar front, the bedload rate significantly decreased, illustrating the lee effect that is materialized by a decrease in bedload sediment transport (Fig. 10a). This, showed that the hydrophone is sensitive enough to detect this local phenomenon induced by the presence of a bar immediately upstream. The bedload rates range from about 8 $g.s^{-1}.m^{-1}$ on the bar crest to 376 $g.s^{-1}.m^{-1}$ in the bar trough (1 $10^{12}$ $\mu Pa^2$ to 1.7 $10^{14}$ $\mu Pa^2$ of acoustic power, respectively). According to flow velocity measurements, it appears that a 2 m high bar front can influence flow velocity and bedload transport rates up to the reattachment point located approximately 100 m downstream. Downstream the bar front, the bedload transport rate increased from 11h06min (Fig. 10a) that would be in coincidence with the flow reattachment point. Further downstream, the bedload transport rate increased from 8.5 to 23.4 $g.s^{-1}.m^{-1}$ (representing respectively an acoustic power of $1.2 \times 10^{12}$ $\mu Pa^2$ to $4.1 \times 10^{12}$ $\mu Pa^2$), where dunes exhibit a more regular shape increasing their amplitudes from 0.02 m to 0.4 m, approximately. In the left part of the channel (Fig. 10b), the drift was located on the stoss side of a bar where larger dunes were observed (about 1 m in height) with





superimposed small dunes (height approximately equal to 0.3 m). The bedload transport rate calculated above
these bedforms increased near the crests of the large dunes (about 80 g.s$^{-1}$.m$^{-1}$) and decreased in the troughs
(about 50 g.s$^{-1}$.m$^{-1}$) where superimposed bedforms were smaller (Fig. 10b).

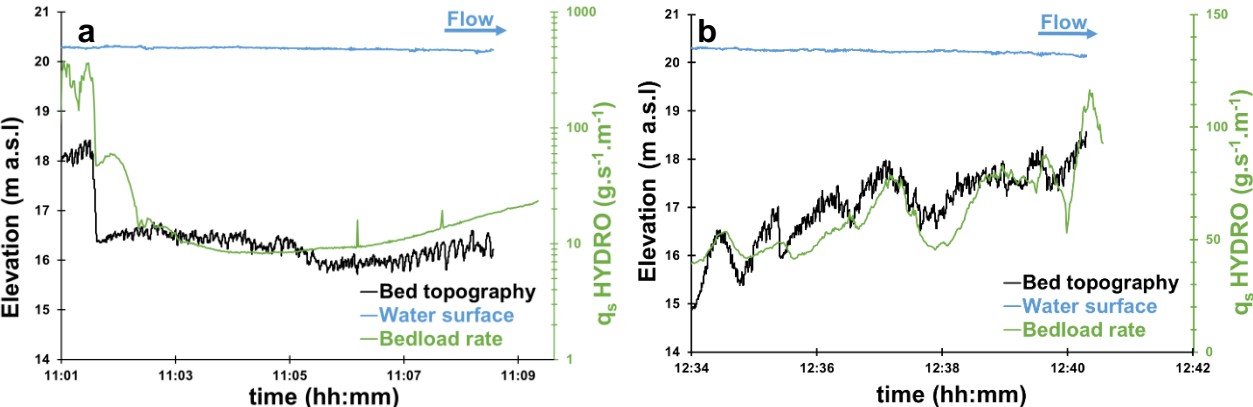


**Fig. 10: Bedload rates calculated on bedforms using the hydrophone and Eq. (15) near a bar front (a) and on a dune**
**field (b). Bed topography and water surface along two longitudinal bathymetric profiles for the 08/02/2018 survey,**
**Q=1550 m$^3$.s$^{-1}$: (a), P10, mean water depth = 3.8 m. The profile length from 11:01 to 11:09 corresponds to 400 m; (b),**
**P12, mean water depth = 3.4 m. The profile length from 12:34 to 12:41 corresponds to 518 m.**
**5. Discussion**
**5.1. Relevance of acoustics for computing bedload transport rates**
Despite their lack of accuracy and their low spatial representativeness, isokinetic samplers allow a direct
measurement of bedload and represents. To this date, measurement based on isokinetic samplers is the only
technique used to compare or calibrate another bedload sediment gauging method in large rivers. The presence
of bars affect sediment transport locally and make sampling method very sensitive to the location of the sampling
point. For low water discharge (below mean annual discharge, 800 m$^3$.s$^{-1}$), bars are emerged and reduce
considerably the width where sediment transport occurs. The number of sampling points decrease with discharge
(because bars were not flooded) leading to a higher bedload rates variability (Fig. 6b). Moreover, for these hydraulic
conditions, bedload transport occurs over a very low thickness reducing the efficiency of the sampler (initially
calibrated to 50%, Eijkelkamp, 2003). The presence of dunes influences the performance of the sampler by
preventing the exact positioning of sampler mouth on the river bed.
The use of hydrophones to estimate bedload transport in a lowland sandy gravel-bed river constitutes a new
research topic. As discussed by several authors, the use of hydrophones was so far restrained to gravel-bed rivers
(Bedeus and Ivicsics, 1963; Barton et al., 2010; Hilldale et al., 2014; Thorne, 2014; Marineau et al., 2016; Geay et
al., 2017) or marine environments (Thorne et al., 1984; Thorne, 1986; Blanpain et al., 2015). More recently, Geay
et al. (2020) highlighted that the acoustic power measured with a hydrophone can be correlated with isokinetic





sampler measurements of bedload in fluvial environments characterized by bed slopes varying between 0.05 and
2.5% and channel width ranging between 8 and 60 m. In these mountainous environments, the median grain size
ranged between 0.9 and 62 mm (n=582 samples). In our study, the downstream reach of the Loire River shows
smaller slope (S=0.02%), a wider channel (W=500 m), and a median grain size ranging between 0.3 mm to 3.1
mm (n=450 samples). The hydrophone is an efficient tool for sediment transport gauging, allowing the
measurement of numerous sampling points (average of 17 sampling points) during few time (an hour). This high
spatial discretization makes the hydrophone functional over a wide range of discharges (even for low water
discharge, Fig. 6b) by catching the high spatial variability of bedload transport. It should be pointed that the
regression calculated in the present study (Eq. 15) is obtained from unit bedload rates (from several samples) and
the acoustic power resulting to a unique acoustical drift, whereas Geay et al. (2020) compared averaged cross
section bedload rates and acoustic power. Despite these differences, the data presented above corroborate the
results by Geay et al. (2020) and support their conclusions concerning the determination of a global calibration
curve between acoustic power and bedload rates by extending its application to the lowland sandy gravel-bed
rivers. Although this need to be confirmed by further investigations to better understand parameters that control the
acoustic power measured (such as the propagation of sound waves in water (Geay et al., 2019) and their
attenuation, the saltation length and associated impact celerity, or sediment grain size), results presented in this
study suggest that the hydrophone method could be an efficient way to measure and to map bedload transport
rates in a wider range of fluvial systems.
Several laboratory studies have been carried out (Ramooz and Rennie, 2010; Conevski, 2019) and rivers
instrumented with aDcp to determine bedload rates (Rennie et al., 2002; Rennie and Millar, 2004; Gaeuman and
Jacobson, 2006; Gaeuman and Pittman, 2010; Brasington et al., 2011). This method remains site-specific and
there is no general agreement between bedload rates and apparent velocity (Rennie and Villard, 2004). The
response of aDcp to bedload transport depends on the frequency of the device used and grain size (Gaeuman and
Rennie, 2006) with a strong influence of near-bed suspended sediments in sandy environment (Rennie et al.,
2002). Moreover, the accuracy of the measurement on a single cross section depends on the water depth
heterogeneity that influences the bottom track sampling area of beams and make the aDcp method location
sensitive when bedforms are present (Fig. 8b). When negative values of apparent bedload velocity are measured,
the value is considered as null and interpolated over a width that is probably wider than the effective width where
bedload transport is null. In consequence, total bedload transport rates estimated with Eq. (14) lead to an
underestimation of BTMA bedload transport rates, especially for low water discharge (Fig. 6b). Estimation of
bedload rates using empirical equations is limited by the number of variables that are difficult to measure in the
field (e.g. thickness and concentration of active layer, Conevski et al., 2018; Holmes, 2010; Kostaschuck et al.,
2005; Latosinski et al., 2017; Villard et al., 2005). Results of Fig. 3b suggest that the apparent bedload velocity
measured by aDcp is the velocity of a sediment layer of about $1D_{50}$ thick, defined by Church and Haschenburger





(2017) as the "dynamic active layer". This result is in agreement with observations made during this study with
video recorders for low flow conditions but it can be criticized for high flow conditions in a sandy-gravel bed river.
The active layer thickness should increase with discharge as more particles are transported as suspension. Van
Rijn (1984) defined the bedload layer thickness equal to the saltation height. Bedload transport rates calculated
using Eq. (5) and Eq. (6) were overestimated in comparison of those measured with BTMA. If we consider that the
thickness of active layer is underestimated for these hydraulic conditions in the Eq. (4) (equal to $D_{50}$), this suggests
that the apparent bedload velocity could be overestimated by aDcp when sediments are in suspension near the
bed (water bias) and the aDcp frequency is too high (M9, 3 MHz). In this study, the high aDcp frequency seems to
compensate the underestimation of bedload layer thickness in the Eq. (4) by measuring higher apparent velocity
of particles in saltation/suspension. In the case of the use of Eq. (5) and Eq. (6), the aDcp frequency should be
adapted with hydraulic conditions and sediment grain size, decreasing the frequency for flood events.
Contrarily to the aDcp, the DTM allows the investigation of "event active layer" (Church and Haschenburger, 2017).
The DTM is not a punctual measurement of bedload. Consequently, in presence of macroforms such as bars, it is
difficult to compare with BTMA samples because it takes into account dunes that are not necessarily present at the
BTMA sampling point (typically downstream a bar lee side). To some extent, the DTM and BTMA methods integrate
bedload longitudinally at different scales. The presence of a local disturbance (or migrating bedform at low celerity)
will affect the measurement. The determination of dune celerity by post-processing is time-consuming compared
with the determination of dune morphology and the existing open access post-processing tools. In order to
determine bedload rates with empirical equations, this method needs a calibration coefficient that is difficult to
measure in field studies (Ten Brinke et al., 1999; Wilbers, 2004). Nevertheless, DTM remains an accurate method
to estimate bedload transport in the Loire River (Fig. 6b) where dunes are present and high enough (over the mean
annual discharge).
As suggested by previous authors, both aDcp (Kenney, 2006) and hydrophone (Bedeus and Ivicsics, 1963) allow
a reliable representation of bedload fluxes on a cross section through the regressions with bedload rates obtained
using samplers. Fig. 8a and Fig. 8b highlight the benefits of the use of acoustics devices for the determination of
bedload transport rates in a large sandy gravel-bed rivers. In the present study, the time needed in the field to
complete the red, yellow, blue and black lines of Fig. 8b (BTMA, DTM, aDcp and hydrophone methods,
respectively) are about 1 day, 4 hours, 1.5 hours and 45 minutes, respectively. This underlines the high potential
of hydrophones to quantify bedload in large rivers with high spatial variability of sediment transport and map
bedload sediment fluxes at a large scale as proposed by Williams et al. (2015) using the aDcp. Moreover, all indirect
methods tested here seem to be able to quantify total bedload transport as efficient as the direct method (Fig. 6b)
but special care should be taken to local estimation of bedload rates (Fig. 8a and Fig. 8b).
Finally, regarding the correlation of aDcp and hydrophone with BTMA (Fig. 3a and Fig. 5a), we can raise the
question of the reference method. Indeed, the regression between aDcp and hydrophone is more significant





(R²=0.76) and it could be the quality and the accuracy of BTMA sampling that reduce the quality of indirect
measurement regressions.

**5.2. Hydrophone and aDcp sensibility to bedform observations**

Passive (hydrophone) and active (aDcp) acoustic devices are rarely used for analysis of bedload transport rates
associated with bedforms in relatively large lowland rivers. Several studies mention differences in apparent bedload
velocity according to the location on bedforms (Rennie and Millar, 2004; Villard and Church, 2005; Gaeuman and
Jacobson, 2006; Holmes, 2010; Latosinski et al., 2017). These authors have shown that apparent bedload velocity
increases from trough to crest of the dune and confirmed previous observations made with samplers (Kostachuck
and Villard, 1996; Carling et al., 2000). These observations were made on large dunes that migrate too slowly to
allow a continuous measurement along bedform. Our study completes these observations offering a fixed and
continuous measurement of apparent bedload velocity and providing bedload transport rates estimation based on
a calibration curve. The mean time between two subsequent crests (1 hour) shows that even for small bedforms
($H_D$ = 0.05 to 0.2 m, Fig. 9a and Fig. 9b), the aDcp location significantly influences the bedload rates calculated
over a dune field (0.03 to 0.08 m.s-1 of difference between crest and trough). This suggests that care should be
taken using this method on river beds where large dunes are present but also when small dunes are migrating.
According to Rennie and Millar (2004), the sampling area diameter increases with flow depth and is more or less
equal to flow depth. Our protocol minimizes flow depth by submerging the aDcp and therefore minimizes the beams
sampling diameter, hence, minimizes the probability to sample stoss or lee sides of the same dune simultaneously.
In our study context, the acoustic power recorded by the hydrophone was not affected by the distance between the
hydrophone and the river bed. To our knowledge, there are no references mentioning investigations on bedload
transport rates associated with bedforms using a hydrophone. At a large time step (mean aDcp and hydrophone
samples), the apparent bedload velocity and the acoustic power did not follow the observed trend of mean bedform
characteristics derived from DTM measurement (dune celerity and dune height). This could be explained by the
difference of spatial scales between DTM and other methods. For a finer time step, our results showed that acoustic
power is able to describe the influence of bars on bedload sediment transport (Fig. 10a). Moreover, as for the aDcp,
the hydrophone also detects the theoretical pattern of bedload transport rates associated with dune migration. As
shown by Reesink et al. (2014), the presence of bars influences the development of dunes downstream and the
distance between bar crest and dune initiation point increases with flow velocity. Specifically, the hydrophone is
able to record an increasing acoustic power assumed to be associated with the increasing dune height downstream
of the bedform initiation point (about 11h06, Fig. 10a). In the present study, dunes smaller than 0.4 m (Fig. 10a)
were not high enough to allow the observation of changes in the acoustic power along the bedform stoss sides.
Conversely, for higher dunes ($H_D$ = 1 m, Fig. 10b) the bedload generated noise can be well recorded by the
hydrophone.



Hydrophone lower detection limit was not reached during our study whereas the dispersion of bedload rates
measured with samplers for low apparent bedload velocity suggests that the lower detection limit of the apparent
bedload velocity by the aDcp seems to be about 1 cm.s$^{-1}$ (Rennie et al., 2017). This lower detection limit of the
apparent bedload velocity should be reduced to the bottom track uncertainty by using our protocol with a
submerged and fixed aDcp device.
**6. Conclusions**
In this work, direct (BTMA isokinetic samplers) with active (aDcp and DTM) and passive (hydrophone) acoustic
measurements of bedload transport rates were compared in a large, sandy-gravel bed river characterized by the
presence of bars and superimposed dunes. Calibration curves between apparent bedload velocity measured using
aDcp and bedload rates measured using BTMA samplers were established but remain site-specific and strongly
correlated to grain size. DTM seemed to be inappropriate where macroforms are present, as it influences the
location and the size of superimposed mesoforms. The calculation of bedload rates with empirical formulas is
sensitive to bedload discharge coefficient for DTM and to thickness and concentration of active layer for aDcp.
These parameters remain always difficult to measure in the field. The use of the hydrophone to monitor bedload
transport rates is for the moment mainly limited to gravel-bed rivers. Results presented in this study highlight the
potential of this technique for the quantification and mapping of bedload transport rates in relatively large river
channels where migrating bedforms are present. This study consolidates a previous recent study (Geay at al.,
2020) by extending a general calibration curve to large sandy-gravel bed rivers. The hydrophone global calibration
curve allows a good representation of the bedload fluxes evolution through a cross section. The method is cheaper
to implement and more efficient than the reference method. This might allow mapping bedload transport rates by
interpolating acoustic power along several cross sections performed on a large sandy gravel bed river. Moreover,
acoustic devices (aDcp and hydrophone) are able to catch the evolution of bedload signal along bedforms stoss
and lee sides with some limitation of bedform size for the hydrophone and signal noise for the aDcp. Regarding
results of the comparison between bedload velocity and acoustic power, the association of aDcp and hydrophone
could be an efficient way to control the quality of both devices. However, additional measurements need to be done
to explore the quality of the regression in other river environments (different grain sizes, river-bed slope or
propagation effect). Finally, the lack of post-processing open access tools for these surrogate technologies slow
the development and use of these devices to bedload rates determination.



## Appendices

Appendix A: Hydrophone dataset

| Date | Number of Hydrophone Drifts | average drift duration (s) | mean acoustic power (Pa²) |
|---|---|---|---|
| 08/02/2018 | 24 | 60 | 2.17E+13 |
| 17/05/2018 | 24 | 80 | 1.46E+12 |
| 15/04/2019 | 11 | 37 | 1.66E+12 |
| 16/04/2019 | 11 | 42 | 2.25E+12 |
| 17/04/2019 | 11 | 28 | 1.42E+12 |
| 18/04/2019 | 11 | 30 | 2.35E+12 |
| 27/05/2019 | 8 | 42 | 5.07E+11 |
| 29/05/2019 | 9 | 36 | 2.00E+12 |
| 09/12/2019 | 22 | 29 | 6.67E+12 |
| 10/12/2019 | 21 | 22 | 7.69E+12 |
| 11/12/2019 | 22 | 27 | 8.84E+12 |
| 12/12/2019 | 13 | 27 | 8.97E+12 |
| 19/12/2019 | 22 | 25 | 2.41E+13 |
| 18/05/2020 | 8 | 50 | 4.53E+12 |
| 19/05/2020 | 8 | 30 | 3.82E+12 |
| 20/05/2020 | 17 | 36 | 3.07E+12 |





Appendix B: ADcp dataset

| Date | Number of aDcp sampling points | aDcp frequency (kHz) | aDcp type | Average aDcp sampling duration (s) | mean Va (m.s⁻¹) | mean water depth (m) | mean flow velocity (m.s⁻¹) |
|---|---|---|---|---|---|---|---|
| 27/03/2017 | 4 | 1200 | RG | 3909 | 0.013 | 2.0 | 0.7 |
| 28/03/2017 | 4 | 1200 | RG | 3279 | 0.015 | 2.1 | 0.7 |
| 29/03/2017 | 4 | 1200 | RG | 3276 | 0.011 | 2.2 | 0.7 |
| 30/03/2017 | 4 | 1200 | RG | 1707 | 0.009 | 2.1 | 0.8 |
| 15/05/2017 | 3 | 1200 | RG | 3018 | 0.002 | 1.3 | 0.8 |
| 16/05/2017 | 2 | 1200 | RG | 2315 | 0.010 | 1.0 | 0.8 |
| 17/05/2017 | 3 | 1200 | RG | 2618 | 0.003 | 1.4 | 0.8 |
| 18/05/2017 | 3 | 1200 | RG | 2467 | 0.002 | 1.6 | 0.8 |
| 04/12/2017 | 3 | 1200 | RG | 2647 | 0.000 | 1.2 | 0.7 |
| 05/12/2017 | 3 | 1200 | RG | 2657 | 0.008 | 1.2 | 0.6 |
| 06/12/2017 | 3 | 1200 | RG | 2246 | 0.000 | 1.2 | 0.7 |
| 07/12/2017 | 3 | 1200 | RG | 2588 | 0.002 | 1.3 | 0.7 |
| 08/12/2017 | 3 | 1200 | RG | 3400 | 0.003 | 1.2 | 0.6 |
| 15/01/2018 | 3 | 1200 | RG | 3256 | 0.084 | 3.2 | 1.1 |



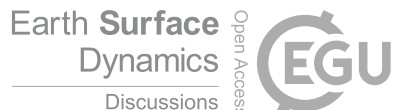

| | | | | | | | |
|---|---|---|---|---|---|---|---|
| 16/01/2018 | 3 | 1200 | RG | 1800 | 0.058 | 2.9 | 1.0 |
| 17/01/2018 | 4 | 1200 | RG | 3185 | 0.041 | 2.7 | 1.0 |
| 18/01/2018 | 4 | 1200 | RG | 3656 | 0.055 | 2.8 | 1.0 |
| 19/01/2018 | 3 | 1200 | RG | 2029 | 0.075 | 2.7 | 1.1 |
| 30/01/2018 | 3 | 1200 | RG | 2138 | 0.051 | 3.9 | 1.1 |
| 31/01/2018 | 3 | 1200 | RG | 2056 | 0.070 | 3.7 | 1.1 |
| 08/02/2018 | 4 | 3000 | M9 | 1136 | 0.038 | 2.8 | 0.9 |
| 14/05/2018 | 4 | 3000 | M9 | 2130 | 0.002 | 1.2 | 0.6 |
| 15/05/2018 | 4 | variable | M9 | 1133 | 0.011 | 1.5 | 0.6 |
| 16/05/2018 | 3 | variable | M9 | 948 | 0.002 | 1.4 | 0.7 |
| 17/05/2018 | 3 | 1200 | RG | 1346 | 0.003 | 1.7 | 0.7 |
| 15/04/2019 | 3 | variable | M9 | 2601 | 0.009 | 1.2 | 0.8 |
| 16/04/2019 | 3 | 3000 | M9 | 1687 | 0.006 | 1.1 | 0.7 |
| 17/04/2019 | 3 | variable | M9 | 1152 | 0.010 | 1.0 | 0.7 |
| 18/04/2019 | 3 | variable | M9 | 3580 | 0.008 | 0.9 | 0.7 |
| 27/05/2019 | 1 | 3000 | M9 | 10949 | 0.003 | 0.9 | 0.8 |





| | | | | | | | |
|---|---|---|---|---|---|---|---|
| 29/05/2019 | 1 | 3000 | M9 | 11539 | 0.029 | 0.9 | 0.7 |
| 09/12/2019 | 2 | 3000 | M9 | 1753 | 0.023 | 1.7 | 0.8 |
| 10/12/2019 | 3 | 3000 | M9 | 1160 | 0.018 | 2.1 | 0.8 |
| 11/12/2019 | 3 | 3000 | M9 | 1288 | 0.027 | 1.6 | 0.9 |
| 12/12/2019 | 2 | 3000 | M9 | 1349 | 0.032 | 2.1 | 0.8 |
| 19/12/2019 | 5 | 3000 | M9 | 1221 | 0.056 | 3.0 | 1.1 |
| 19/05/2020 | 2 | 3000 | M9 | 7318 | 0.014 | 1.0 | 0.7 |
| 20/05/2020 | 4 | 3000 | M9 | 2988 | 0.004 | 1.6 | 0.7 |



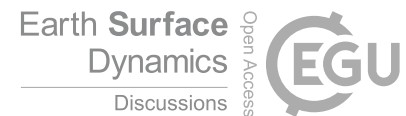

Appendix C: BTMA dataset

| Date | Discharge | Measurements type | Number of BTMA sampling points | Number of BTMA samples | Mean unit bedload rate | D50 | D90 |
|---|---|---|---|---|---|---|---|
| | (m³.s⁻¹) | | | | (g.s⁻¹.m⁻¹) | (mm) | (mm) |
| 28/11/2016 | 1420 | BTMA & DTM | 3 | 50 | 38.1 | 0.8 | 3.0 |
| 29/11/2016 | 1460 | BTMA & DTM | 4 | 79 | 31.5 | 0.9 | 3.5 |
| 30/11/2016 | 1300 | BTMA & DTM | 4 | 80 | 33.2 | 0.8 | 2.9 |
| 01/12/2016 | 1100 | BTMA & DTM | 4 | 79 | 32.2 | 0.8 | 2.6 |
| 27/03/2017 | 687 | BTMA. aDcp & DTM | 4 | 80 | 25.3 | 0.7 | 2.9 |
| 28/03/2017 | 752 | BTMA. aDcp & DTM | 4 | 80 | 28.5 | 0.8 | 3.0 |
| 29/03/2017 | 827 | BTMA. aDcp & DTM | 4 | 57 | 29.0 | 0.8 | 3.8 |
| 30/03/2017 | 812 | BTMA. aDcp & DTM | 4 | 80 | 19.3 | 0.8 | 3.8 |
| 15/05/2017 | 346 | BTMA. aDcp & DTM | 3 | 60 | 6.3 | 0.9 | 4.8 |
| 16/05/2017 | 354 | BTMA. aDcp & DTM | 3 | 60 | 13.5 | 0.8 | 5.0 |
| 17/05/2017 | 401 | BTMA. aDcp & DTM | 3 | 55 | 9.0 | 0.9 | 4.7 |
| 18/05/2017 | 447 | BTMA. aDcp & DTM | 3 | 60 | 1.9 | 1.2 | 7.0 |
| 04/12/2017 | 243 | BTMA & aDcp | 3 | 60 | 1.8 | 1.1 | 7.4 |



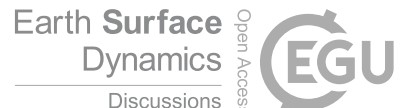

| | | | | | | | |
|---|---|---|---|---|---|---|---|
| 05/12/2017 | 241 | BTMA. aDcp & DTM | 3 | 60 | 3.7 | 1.0 | 8.6 |
| 06/12/2017 | 243 | BTMA. aDcp & DTM | 3 | 60 | 6.6 | 1.2 | 6.7 |
| 07/12/2017 | 246 | BTMA. aDcp & DTM | 3 | 60 | 5.1 | 1.2 | 5.1 |
| 08/12/2017 | 226 | BTMA. aDcp & DTM | 3 | 60 | 5.0 | 1.6 | 7.9 |
| 15/01/2018 | 1740 | BTMA. aDcp & DTM | 3 | 60 | 61.4 | 1.0 | 2.9 |
| 16/01/2018 | 1550 | BTMA. aDcp & DTM | 3 | 60 | 89.4 | 0.9 | 2.8 |
| 17/01/2018 | 1460 | BTMA. aDcp & DTM | 4 | 80 | 53.2 | 0.8 | 3.0 |
| 18/01/2018 | 1540 | BTMA. aDcp & DTM | 4 | 80 | 97.7 | 1.0 | 3.3 |
| 19/01/2018 | 1510 | BTMA. aDcp & DTM | 3 | 60 | 55.6 | 0.8 | 2.6 |
| 30/01/2018 | 2410 | BTMA. aDcp & DTM | 3 | 60 | 68.6 | 0.8 | 2.3 |
| 31/01/2018 | 2290 | BTMA. aDcp & DTM | 3 | 59 | 55.8 | 0.8 | 2.2 |
| 08/02/2018 | 1550 | BTMA. aDcp. DTM. Hydrophone | 4 | 69 | 63.4 | 0.8 | 2.5 |
| 14/05/2018 | 443 | BTMA. aDcp & DTM | 4 | 79 | 2.2 | 0.9 | 2.7 |
| 15/05/2018 | 449 | BTMA & aDcp | 4 | 79 | 2.5 | 1.1 | 3.2 |
| 16/05/2018 | 547 | BTMA. aDcp & DTM | 3 | 60 | 6.6 | 1.2 | 4.4 |
| 17/05/2018 | 604 | BTMA. aDcp. DTM. Hydrophone | 3 | 60 | 7.2 | 1.2 | 4.4 |



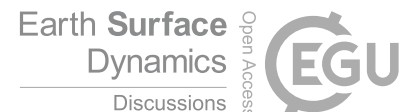

| | | | | | | | |
|---|---|---|---|---|---|---|---|
| 15/04/2019 | 253 | BTMA. aDcp & Hydrophone | 3 | 60 | 22.1 | 0.9 | 3.3 |
| 16/04/2019 | 243 | BTMA. aDcp & Hydrophone | 3 | 60 | 22.1 | 1.1 | 5.1 |
| 17/04/2019 | 240 | BTMA. aDcp & Hydrophone | 3 | 60 | 24.9 | 1.2 | 3.7 |
| 18/04/2019 | 238 | BTMA. aDcp & Hydrophone | 3 | 58 | 16.4 | 1.0 | 5.3 |
| 27/05/2019 | 225 | BTMA. aDcp. DTM. Hydrophone | 1 | 26 | 34.6 | 1.0 | 4.8 |
| 29/05/2019 | 210 | BTMA. aDcp. DTM. Hydrophone | 1 | 28 | 22.0 | 1.1 | 3.3 |
| 09/12/2019 | 944 | BTMA. aDcp. DTM. Hydrophone | 2 | 40 | 29.1 | 0.7 | 2.5 |
| 10/12/2019 | 898 | BTMA. aDcp. DTM. Hydrophone | 3 | 60 | 20.1 | 0.6 | 2.5 |
| 11/12/2019 | 923 | BTMA. aDcp. DTM. Hydrophone | 3 | 45 | 34.9 | 0.8 | 2.4 |
| 12/12/2019 | 925 | BTMA. aDcp. DTM. Hydrophone | 2 | 37 | 26.4 | 0.7 | 2.7 |
| 19/12/2019 | 2050 | BTMA. aDcp. DTM. Hydrophone | 5 | 50 | 58.8 | 0.9 | 3.4 |
| 18/05/2020 | 514 | BTMA & Hydrophone | 1 | 57 | 19.7 | 0.9 | 2.8 |
| 19/05/2020 | 500 | BTMA. aDcp & Hydrophone | 2 | 79 | 30.9 | 1.0 | 2.6 |
| 20/05/2020 | 470 | BTMA. aDcp & Hydrophone | 4 | 40 | 14.5 | - | - |





Appendix D: DTM dataset

| Date | Number of DTM profiles | average interval time DTM (min) | Number of dunes | Mean $H_D$ (m) | Mean $L_D$ (m) | Mean $C_D$ (m.d$^{-1}$) |
|---|---|---|---|---|---|---|
| 28/11/2016 | 2 | 18 | 65 | 0.19 | 2.88 | 43.0 |
| 29/11/2016 | 3 | 20 | 168 | 0.22 | 3.69 | 34.8 |
| 30/11/2016 | 3 | 18 | 121 | 0.24 | 4.16 | 37.6 |
| 01/12/2016 | 3 | 19 | 104 | 0.25 | 4.69 | 37.6 |
| 27/03/2017 | 3 | 38 | 132 | 0.13 | 3.13 | 28.3 |
| 28/03/2017 | 3 | 44 | 97 | 0.13 | 2.96 | 24.2 |
| 29/03/2017 | 3 | 43 | 117 | 0.14 | 3.25 | 25.7 |
| 30/03/2017 | 3 | 39 | 138 | 0.14 | 3.42 | 28.0 |
| 15/05/2017 | 3 | 65 | 20 | 0.04 | 2.17 | 18.1 |
| 16/05/2017 | 3 | 42 | 11 | 0.05 | 2.02 | 26.7 |
| 17/05/2017 | 3 | 38 | 18 | 0.05 | 2.01 | 28.0 |
| 18/05/2017 | 3 | 28 | 34 | 0.08 | 1.95 | 30.9 |
| 05/12/2017 | 1 | 73 | 48 | 0.13 | 2.90 | 17.9 |
| 06/12/2017 | 1 | 98 | 68 | 0.16 | 3.44 | 14.9 |





| Date | | | | | |
|---|---|---|---|---|---|
| 07/12/2017 | 1 | 72 | 63 | 0.17 | 3.62 | 17.3 |
| 08/12/2017 | 1 | 66 | 69 | 0.19 | 3.95 | 14.8 |
| 15/01/2018 | 6 | 23 | 228 | 0.32 | 6.66 | 38.1 |
| 16/01/2018 | 2 | 28 | 46 | 0.24 | 3.58 | 47.6 |
| 17/01/2018 | 3 | 32 | 52 | 0.25 | 4.36 | 34.9 |
| 18/01/2018 | 3 | 55 | 120 | 0.28 | 5.33 | 28.0 |
| 19/01/2018 | 3 | 31 | 110 | 0.26 | 4.95 | 31.4 |
| 30/01/2018 | 3 | 25 | 103 | 0.32 | 5.75 | 45.3 |
| 31/01/2018 | 4 | 22 | 83 | 0.28 | 5.02 | 45.4 |
| 08/02/2018 | 3 | 60 | 59 | 0.26 | 4.67 | 28.2 |
| 14/05/2018 | 6 | 35 | 58 | 0.06 | 2.92 | 20.8 |
| 16/05/2018 | 4 | 38 | 60 | 0.05 | 1.96 | 18.8 |
| 17/05/2018 | 6 | 34 | 81 | 0.05 | 1.98 | 22.3 |
| 27/05/2019 | 1 | 29 | 3 | 0.03 | 1.40 | 62.7 |
| 29/05/2019 | 1 | 26 | 7 | 0.03 | 1.28 | 30.7 |
| 09/12/2019 | 6 | 49 | 121 | 0.22 | 3.10 | 28.1 |



| | | | | | | |
|---|---|---|---|---|---|---|
| 10/12/2019 | 6 | 42 | 227 | 0.17 | 3.60 | 33.2 |
| 11/12/2019 | 6 | 49 | 254 | 0.16 | 3.46 | 33.1 |
| 12/12/2019 | 6 | 50 | 297 | 0.18 | 3.82 | 35.9 |
| 19/12/2019 | 3 | 44 | 79 | 0.28 | 4.34 | 42.1 |





## Authors contribution

J. Le Guern prepared the manuscript with contributions from all co-authors. T. Geay, A, Hauet, S. Zanker, S. Rodrigues helped to elaborate the experimental protocol. T. Geay developed the hydrophone signal processing tools. A. Duperray P. Jugé, L. Vervynck, A. Hauet, S. Zanker, T. Geay, S. Rodrigues and J. Le Guern took part in surveys. A. Duperray P. Jugé, and L. Vervynck made the bathymetry post-processing. S. Rodrigues, N. Claude and P. Tassi supervised this study.

## Competing interests

The authors declare that they have no conflict of interest.

## Acknowledgement

This study is a part of the Ph.D. thesis of the first author funded by the POI FEDER Loire (Convention no. 2017-EX002207) and Agence de l'Eau Loire Bretagne (decision no.2017C005), conducted in the frame of the Masterplan Plan Loire Grandeur Nature. We thank EDF DTG and ANR Intelligence des Patrimoines Phase 2 for lending us acquisition equipment. Exagone Company is acknowledged for providing us data from Teria network, Voie Navigable de France (VNF) for their logistical support during field campaigns and Polytech Tours. J.-P. Bakyono, P. Berault, T. Bulteau, B. Deleplancouille, Y. Guerez, T. Handfus, I. Pene and C. Wintenberger, are acknowledged for their help during field investigations and grain size analyses. We are grateful to T. Geay and J. Hugueny for the hydrophone treatment and aDcp data post-processing tools, respectively. The authors wish thank Karl Matthias Wantzen for the English translation of the manuscript.

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
