# Peer review of "Relevance of acoustic methods to quantify bedload transport and bedform dynamics in a large sandy-gravel bed river"

_Earth Surface Dynamics, 2020_

## Referee Comment (RC1) · Anonymous Referee #1 · 4 Nov 2020

Evaluation

This is an important study that is based on a comprehensive set of bedload transport measurements obtained from an experiment designed to compare three acoustic methods with isokinetic sampling at locations across a transect of the Loire River during different flow stages. It is significant in that it presents hydrophone-derived measurements in a sandy gravel-bed river, which differentiates it from the numerous studies in gravel-bed rivers, and that it presents hydrophone-derived measurements with a spatial distribution that enables bedload transport to be related to bedform morphology.

The paper addresses relevant scientific questions within the scope of ESurf and

presents novel data with substantial conclusions. The results are sufficient to reach the conclusions given, however some further discussion of the results would improve the quality of the paper.The paper is currently in need of further editing to improve the quality of written English and to more accurately describe the methods in order that other researchers may replicate those methods. The language needs to be more fluent and precise. The overall structure of the paper is relatively clear and a proper consideration is given to related work in setting the novel aspects of the research in proper context. Specifically, in relation to the hydrophone method, I would like to see a clearer explanation of the method and some data that demonstrate the variability of the power-spectral density between hydrophone recordings. I would like the paper to describe if there are significant differences observed between the frequency spectrum of the studied sandy gravel-bed river and the frequency spectra in the comprehensive literature on gravel-bed rivers. How does the different grain size distribution affect the observed frequency spectrum?

Specific comments

Lines 57-58 to estimate the capacity of acoustic signals to detect the bedload axes on relatively wide cross-sections for various discharge conditions

I'm not sure what this means. Are you referring to estimating the accuracy of acoustic methods for measuring bedload transport rates for wide transects across a arrange of flow conditions?

Line 116 giving better results

How do you know that they are better results? Please provide a clearer description of the method used to determine data quality.

Lines 147-148 and could allow a better understanding of the apparent bedload velocity gradient along bedforms.

It's not clear to me what this means. Why would this give a better understanding of

ESurfD
the bedload velocity gradient along bedforms? Is it because the ADCP footprint is relatively small compared with the bedform dimensions? If so, please state typical bedform dimensions relative to the footprint.

Lines 161-162 The determination of a proxy to evaluate sediment transport directly from DTM measurements is difficult.

Explain why it was difficult.

Lines 240-241 and the integration of median PSD over a wider range of frequency in the present study.

What is meant by median? I'm not sure what this is referring to and I think it needs to be explained more clearly. Is the PSD integrated across the median value of each frequency bin?

Lines 251-257 The comparison can be made between indirect methods to discuss the acceptability of the BTMA reference. The apparent bedload velocity and the acoustic power are not well-correlated with mean dune morphological parameters (dune celerity and dune height). The aDcp method is measuring the apparent velocity of the grain being transported from the stoss to the lee side of a dune. It must be noted that apparent bedload velocity is higher than dune celerity with about a factor 100, whereas the grain size (D50) is smaller than dune height with the same order. Therefore, sediments that are 100 times smaller than dune height allows the dune migration with a celerity 100 times smaller than their own celerity.

I'm not sure this section is necessary and I struggle to understand some of the arguments being made. What is meant by the acoustic power and apparent bedload velocity not being well correlated with the morphological parameters? Where is this shown in the data? The comparison between apparent bedload velocity and dune celerity is interesting but I think most readers would already understand that the bedload velocity is much smaller.I'm not sure how this lead on to the next sentence: Lines 257-258 On

the other hand, the apparent bedload velocity is positively correlated with the acoustic power. The RMA regression model explains 76% of the dataset dispersion (Fig. 6a).

What it meant by 'explains 76% of the dataset'? This needs to be accurately described. This paragraph needs to be re-written.

Line 261 and water discharge explain 71% of the dataset dispersion

Again, I do not understand what is meant by explaining the dataset. Please clarify.

Line 286 there was no reference measurements

Why not? – please explain. Can you please add labels, e.g. S1, to the BTMA data points in both panels of Fig. 8 to clearly identify the reference measurements. Why is there no S2 in Fig. 8a? Why are reference measurements missing from Fig. 8b?

Line 294 I would like to see some discussion as to why the hydrophone method is producing larger values of unit bedload rate compared with the BTMA measurements. Just a thought, but could it be related to the omnidirectional hydrophone picking up higher noise magnitudes that are not directly below the boat?

Lines 303-305 For this survey, acoustic signals (i.e. acoustic power, apparent bedload velocity) followed the same evolution pattern as isokinetic samplers along the cross section except for S3.

Again, why is this the case? – is it due to a lack of directionality?

Line 325 Immediately downstream of the bar there are bedload transport values that are higher than those observed further downstream. Is this due to the omnidirectional hydrophone picking up bedload noise from the bar upstream?

Technical Corrections

Lines 52-54 In this work, we compare the efficiency of active (aDcp, echosounder) and passive (hydrophone) acoustic techniques is assessed for the quantification of bedload

transport in a reach of the Loire River (France), which is characterized by the presence of migrating bars and superimposed dunes (Le Guern et al., 2019b).

Please re-write.

Line 127 with D50

Change to 'where D50"

Line 135 Remove double comma

Lines 149-150 a moving windows

Remove 'a'

Line 174 I presume this is a hydrophone with an omnidirectional beam-pattern? If so, please state.

Lines 211-212 a factor 2 of the perfect correlation.

Suggest a change to 'a factor of 2 above and below the perfect correlation'

Line 227 of values in a factor 2, whereas 49% for a discharge coefficient of 0.57.

Suggest a change to: 'of values within a factor of 2 of the perfect correlation compared with 49% of values for a discharge coefficient of 0.57.,'

Line 237 for estimate

Change to: 'to estimate'

Line 265 whereas 73% for the aDcp.

Change to 'compared with 73% for the aDcp'.

Line 278 with respect to bars location

Change to 'with respect to bar location'

Line 286 there was

Change to 'there were'

Line 294 whereas hydrophone overestimates BTMA measurements

Change to 'whereas the hydrophone method overestimates unit bedload rate compared with the BTMA measurements'

Lines 312-313 The amplitude of bedload rates between crest and trough for low flow conditions (Fig. 9a) ranged between 42 g.s-1.m-1 and 69 g.s-1.m-1.

I think this refers to Fig. 9b rather than Fig. 9a. Please check that the plots are correctly referred to in this section.

Line 324 that is materialized by

Change to 'that is characterised by'

Line 325 This, showed that

Change to 'This shows that'

Line 365 during few time (an hour).

Change to 'during a relatively shorter time period (an hour).'

Line 392 Results of Fig. 3b

Change to 'The results shown in Fig. 3b'

Line 395 it can be criticized for

I'm not sure what this means. Please re-write.

Lines 420-421 the red, yellow, blue and black lines of Fig. 8b (BTMA, DTM, aDcp and hydrophone methods, 421 respectively)

Change to the BTMA, DTM, aDcp and hydrophone methods (respectively the red, yellow, blue and black lines of Fig. 8b)

Line 425 taken to local

Change to 'taken with local'

────────────────────

---

## Referee Comment (RC2) · Anonymous Referee #2 · 11 Dec 2020

Referee #2

**General Evaluation**

This manuscript represents a  field investigation of three different acoustic-based methods used to quantify the bedload transport rate over the single cross-section. It is admiring how carefully the experiments were performed and the amount of collected data is rather unique.  The reading of the manuscript was insightful and brought some details into my attention that I was not aware of before, mostly regarding the passive acoustic techniques. I read the manuscript with great interests, therefore quite a few comments and discussion issues are presented in this review. The relation between the bedload dynamics and the measurement capabilities of each technique is well-addressed and by that, this article represents its fair novelty in this research field.  The presented results are clear and presented in fair quality figures and the overall structure is satisfying. Besides, some appendices offer an overview of the available instrumentation and the average measured data.

Yet, it must be noted that the discussions, conclusions, and some of the methods used are not up-to-date and deserve some more thorough investigation. Obviously, the later does not refer to the hydrophone technique, but mostly to the adcps and the physical bedload sampler.

The language and the sentence structure is simple and understandable but the text, in general, deserves a brush-up.  Honestly, I did not focus on that so please re-read and pay more attention to the wording.

I will write my review by giving general comments that address the different techniques and the discussion of the results. Later in the specific comment section, some less important issues will be listed, sometimes referring to the general comments (g.c).

Finally, I would like to apologize for eventual missing citations, some language unclarity or partial information. I wrote this review as quickly as I could.  Please keep in mind that the paper could be published with only minor revisions, specifically in the part of the discussion and not all my comments have to be considered. However, I do think it will necessarily increase the quality of the paper.

**General Comments**

**1. Study site**

Although there are nice tables in the appendices I am missing a Table in the main text with the general hydraulic conditions and sediment characteristics.

a) In the text, there is no info about the **standard deviation of the particle's sizes** of the bedload material. I would also like to see the range of the **particle Reynolds number (Rep)**, the **shear velocity or stress and the critical shear velocity or stress**. It could be also interesting if the authors give information about the reference shear stress at the study site (Parker et. al 1982). Note that this is similar to the critical shear stress but not the same.

b) There are quite a few comments about the measurement capability of the instruments related to the different PSD of the bedload particles, but not details about the PSD of the sampled bedload surface.

c) I have another question about the boat/floating structure given ion Fig2b:

How was It fixed and how intensive was the drifting that you mention several times (in terms of meters)?

**2. BTMA**

A large uncertainty is associated with these physical samplers (Frings, et al., 2017). All physical sampling techniques suffer from difficulties and problems during the measuring procedure, such as: i.) disturbance of the bed; ii.) gap effect (sampler mouth is not in contact with the bed) iii.) sampling time; iv.) positioning above bedforms; v.)loss of material during sampling and raising the sampler, vi.) clogging of the mesh; etc. (Gaweesh, et al., 1994). Concerning these disadvantages, I have several questions:

a) I was wondering how this calibration coefficient (α=2) was adapted. Does it take in consideration all the factors numbers above(check (Banhold, et al., 2016) for detailed analysis of a very similar sampler in Germany) ? It would be good to see a sentence regarding this issue. Calib coefficient of means that the samples underestimate 100 % of the real sediment transport.

b) Is there a defined procedure for sampling? How many samples are sufficient to minimize the uncertainty (Frings, et al., 2017)?

c) Based on which tests? I could not find the citation online…Boiten 2003? It looks like lecture notes, so inside it must be stated how these coefficients are defined and related to another study.

d) And if some experiments exist, were those conducted in similar conditions as your study site?

e) What are the conditions when the sampler would malfunction (e.g. weak transport conditions or too abundant)? There are many examples of low transport rates could be underestimated or overestimated by the physical samplers. Although most of the literature suggests that in weak transport conditions the samplers would over-estimate (e.g., due to digging in the bed), sometimes in sandy rivers, we have noticed many samples to under-sample the total transport (observing videos and total mass caught).

f) Given the Eq. 1 I was wondering if you have done PSD and weight/sieving analysis in the laboratory for each sample? If so, why not using only the dry mass?

g) beside the sieving, there is no information on how you have measured the other parameters, such as the volume V.

h) In the appendices, you give information on how many samples are averaged in the given value. But there is no explanation of how the samples from the two BTMA are used in the processing. Please give information about that.

This information is crucial because it correlates directly with some of the conclusions regarding the measurement capacities of acoustic instruments.

**3. ADCPs**

In general, the methodology and the conclusions given for the ADCPs, as active sonars are not up-to-date and some of the conclusions are partially correct. Here are some comments:

a) The **post-processing procedure** only partially follows some of the latest findings given in (Conevski, et al., 2019) and (Conevski, et al., 2020). Although the authors tried to implement some of the procedure there are some miss-steps. The **de-spiking and the filtering procedure** (discarding the raw BT velocities that are in the opposite direction of the flow) given in these two studies improved the correlation with the samples and gave more realistic velocities. It

seems that the filtering procedure automatically excluded the negative velocities the occur in the so-called recirculation zone of the dunes (which you also mention). In addition, there is some sort of final filtering the involves analysis of the difference of the depth values registered by each beam, which helped to discard some samples that suffered from beam inhomogeneity (Conevski, et al., 2020).

b) The **projection to the water direction** (Wdir GPS) is not clear to me. How these two are defined do you use compass or GPS heading for Wdir and compass heading for Va? Why is this necessary?

You say "To avoid compass and GPS issues, and to eliminate the effect of residual lateral displacement", but then you use again: wdir GPS (GPS ref coordinate system)-bdir BT (compass heading EH). By doing this some of the issues given above are inherited again in the velocity. Also, it is not true that the bedload velocity should align with the water velocity especially in presence of bedforms. If one would like to project one velocity vector on another both should have then the same reference coordinate system or to contain information (rotation and /or translation) about the different reference coordinate system. As far as I understand you use GPS-ENU and Compass – ENU. Note that you are in a fixed position and you do not need water velocity referenced to the BT. Another thing is that if you project over the water velocity the real direction of the bedload is lost, and the final value decreased.

Please correct me if I understood wrong.

c) The concept of volume and surface scattering seems not to be considered in conceptualizing the discussions. Please check (Urick, 1983), also (Conevski, et al., 2019) , characteristics (Conevski, et al., (2020, accepted)). Note that the scattering of the surface formed by the immobile particles below the active layer has large impedance and therefore the echo coming from this scattering is dominant. The roughness of this surface could induce false velocity estimation. This error depends on the acoustic parameters of the instrument and the signal processing algorithm. Please have in mind that this is a complex two-phase (surface +volume) scattering process occurring between two scattering regions (Rayleigh + geometric), so it cannot be easily described.

d) In the same manner, the concept of internal processing and signal processing is not well addressed. Note that the RDI uses broadband codded pulse technology while M9 uses the pulse-coherent method with some internal modifications such as the use of HD smart pulse. Note that the RDIs BB is able to use much finer resolution systems (Brumley, et al., 1991) in the detection of the bottom signal, thus it should be able to define the better immobile bed surface. Note that the main purpose of the bottom tracking signal is to identify the riverbed and it is developed in that way (R. Lee Gordon, 1996). This is one of the reasons why RDI continuously reports lower velocities than M9, regardless of the frequency (Conevski, et al., 2020). This is also related to the acoustic sampling effect described in the same paper.

e) Although it is well known that the grain size is correlated with the backscattering strength even with the bedload particles (Conevski, et al., (2020, accepted); Shiel, 2010), the laboratory investigation (Conevski, et al., 2019; Conevski, et al., (2020, accepted); Conevski, et al., 2020) demonstrated that it is not the only reason for different Doppler / apparent bedload velocities especially in well-developed transport conditions.

f) In the results section, there is no distinction in the Figures between the data measured by the RDI 1200kHz and the M9 (also which frequency). In Latosinski et .al. 2017 and Conevski et al.

2020 it is clearly indicated that different ADCPs give different mean apparent bedload velocities, and it is not strictly dependent on frequency, but also to the acoustic sampling and resolution, pulse length, cell-profiling, signal processing etc. The laboratory tests are given in Conevski et. al 2019 and (Conevski, et al., 2020) also confirm and elaborate on these issues.

g) The timing of 3-10 minutes is enough to get a stable apparent velocity at a single position (Conevski, et al., 2019; Rennie, et al., 2002).

h) What do you mean by site-specific?  This is partially true, and it is only dependent on the shape of the bedforms and the riverbed surface. Some of these elements could be identified and eliminated by filtering procedure, beam inhomogeneity analysis, surveying several cross-sections before positioning. Conevski et. al (2020) reported a relatively high correlation coefficient with data collected at several different positions and two different rivers. As given in the comments above the ADCPs are more instrument dependent than site-specific.

i) The water bias could be assessed by checking the raw water velocity and backscattering echo values from the last available cell, together with the camera data. Also one should expect rather high suspended sediment concentration in the entire profile (above 500 mg/l) to have the effect of water bias (Rennie, et al., 2002; Gaeuman, et al., 2006). Although this effect has not been investigated in laboratory conditions it is expected that the long BT pulse contains more energy and therefore the signal penetrates this suspended layer. Once it penetrates the scattering reflected both,  from the active bedload layer (which has an absolutely higher concentration than the SSC right above) and the surface scattering of the immobile particles below, is absolutely stronger and easier to detect in the autocorrelation as part of Doppler estimation methodology.
What was the SSC concentration in some of the cases? Could you upload a video of the sampling area?

j) The kinematic model.
**Shear velocity**. The semi-empirical approach analytically derived by VanRijn is fairly valid as long as the roughness is assumed to be 3D90.   Since you have the ADCP right above the dune a log-law fitting might effectively correct or the Reynolds method could be also applied (Guerrero, 2011). By obtaining the velocity profile right above the dune stoss side, the drag effect from the bedforms might be negligible, thus the shear stress would be only related to the motion of the grain.  Not sure if this is feasible in your case, but it is an idea.
It seems that the formulas you use for the critical shear velocity do not involve the hiding effect function. Note that you have sand and gravel in the bed mixtures, and therefore a fractional transport.  This might not be crucial for only determining the active layer thickness but it is worth checking because the active layer thickness might increase due to the decrease of the critical shear velocity of the gravel particles, in a presence of sand (Wilcock, et al., 2001; Curran, et al., 2005).
**Porosity**. Which value do you use for the porosity of bedload concentration in the Eq. 4? A constant bedload concentration of 0.65 is utterly false.  Frozen-core bed surface or substrate samples were taken from riverbed have a porosity of 0.4, which simply means the concentration of 0.6. in the same context perfectly, packed particles without consistent shape hardly reach porosity less than 0.3. In the kinematic model, the concentration value is the value of the concentration in the instantaneous (dynamical) bedload active layer. This value shall be at a

maximum lower than 0.6; Van Rijn (1984) reports this as maximum bedload concentration. Rennie and Villard (2004) reported 0.1-0.15 in their model. Note that this is not true for the porosity of the dunes; it is also stated in the paper that you make a distinction between the "dynamical active layer" (I am calling it instantaneous, or max saltation height) and the event - scale exchange layer (e.g., this includes the height of the dunes)

**4.Dune tracking method**

a) The river morphology and the presence of bars (that actively migrate) suggest the dune formation throughout this cross-section would be 3-dimensional, which means that I would not expect straight line dune wavefront. Is it correct?

b) On the other hand, the DTM method that you use is built based on well-developed 2D dunes. Could you please elaborate on this issue? I am afraid that maybe only the dune measurements in the thalweg zone would be most reasonable to use in the comparison with the other two techniques?

c) Why not compare the bedload transport integrated over the entire cross-section from the hydrophones and ADCPs vs the dune tracking of only several longitudinal profiles (e.g., close to the thalweg).

**5.Hydrophones**

a) What is the advantage of drifting of the hydrophones? Isn't there less ambient noise (not bedload flux noise) if they stay fixed? Ok, I read in the Geay et al. (2020) that it is actually the opposite… I did not find a comparison of both cases. How was drifting performed and why the waves hitting the boat or produced by the boat did not affect the measurements?

b) What was the ambient noise level at the presence of the bars? How was the acoustic power correlated with these morphological changes?

c) Was the signal sensitive to the dune's height change?

d) What was the bias of this signal towards the presence of gravel particles?

**6. Final thought**

Finishing up with the reading, I have a feeling that the comparison of the three methods is rather extensive and considering the length of modern scientific papers, it does not permit detailed analysis of each technique. However, this manuscript seems to have more informational character and not a comprehensive analysis of the deployed techniques.

Somehow the DTM method has a slightly different nature of processing and physical explanation. On the other hand, the current paper may involve fewer details about the techniques themselves and focus more on bedload dynamics.

In that sense and maybe in another paper, it could be very interesting to see the correlation between the corrected backscattering strength (see (Conevski, et al., (2020, accepted))) towards the acoustic power from the hydrophones and to focus on only on these two methods, as both represents "local" bedload measurements.

**Specific comments (line by line)**

L21-L23: Please correct the English. Something like: Although the parameters that control the self-generated noise still require analysis…

L23-24 The sentence does not contribute in the abstract and it is not clear. Does it mean that they are not able to measure if bedform migration is not happening?

L29: instead of: It constitutes a prerequisite to the accurate estimation please reformulate in: It is a prerequisite to an accurate estimation…

L: 41 : Please add the latest studies : (Conevski, et al., 2019) (Conevski, et al., (2020, accepted)) (Conevski, et al., 2020)

L 48: Are you talking about multi-beam?

L 64: more info about the PSD (or GSD). See general comment 1.

L 79: "below" substitute with Fig 2.

L 79 : "adopted here as reference". Maybe delete and start new sentence giving information that these data are used as reference data because this method so far is the most reliable… or something similar.

L 79: I suggest using a traditional or conventional method instead of classical.

L92-L100: See general comment 2. all sub comments.

L96: Would you upload a video of the high flood measurements? Thanks.

L102: You say two ADCPs. On Fig 2 I see only one and, in the description, I understood that only one was available per campaign.

L102 "positioned" could be replaced with installed.

L103: Which mode (IC-incoherent or SmartPulse HD?) was used in each case. Please add in the appendices (see, general comment 3 d.).
L120: Please use vector notation and see general comments(g.c.) (3 b.)

I will use g.c. as an abbreviation of general comment.

L124-L130: These are both kinematic models and essentially both describe a unit volume with its specific density moving with unit velocity. The first one is an assumption that the ADCPs are more sensitive to the particles and particle numbers and that by assuming characteristic particle Va would represent

average in all 4 acoustic footprints. Considering g.c. 3c it is rather strange that you get better matching for Eq4 in Fig3b. But then the assumptions for Eq. 5 and the filtering are not correctly applied.

L143-L150: Did you get better results in these surveys, when ADCP was close to the bed?

L150:  Large window like that would do simple de-spike low pass filter. Not the noise but the outliers.
L151: All negative values refer to the average values, if they were negative you discard from the dataset? Or you applied some kind of filtering on the raw data as given in g.c. 3a? If so please cite Conevski et al 2019.
L170: So why did you adopt 0.5 ?

L171: The accuracy of the echo sounding shell be very high. The problem is in the post-processing and identification in the bedforms, isn't it? See g.c 4a.

L174: What are the dimensions of the hydrophone?

L186: I assume the sampling was not synchronized?

L196: Is it the mean of all STDs?

L198: Again. Which of the two BTMA samplers were used in the average value?

L203-204: I am not sure if this comparison makes sense. To me, it looks that your RMA reg model could fairly represent the entire dataset (Rennie + yours, in the range of R2~0.5). Would be interesting to see. Check g.c. 3e.

L210-211: OK now I see that the average negative values were extracted not as in Conevski et al. 2019.

L213: Why did not you calculate for each point separately? What does constant means? For all data? If this is the case then it makes sense eq 4 to perform better.

L219-220> Could you clarify this sentence. Do you mean Eq5 is better than Eq4 or the opposite? Eq.4 does not assume any active layer thickness, but average particle size.

Fig3b.: Do you have something to say about the points down in the middle of the Fig3b?

L221-230:  But that is reasonable. See g.c. 5a-b

L243: Does it involve the bedload concentration or just the velocity?

L254: The comparison is not consistent could you please clarify. The apparent velocity measures the velocity top layer or dynamical active layer, whereas the dune celerity is the mobility of the exchange event active layer, according to Church (2019).

L271: Perhaps, sampling sounds better than gauging.

L290: It is impressive if the DTM method somehow worked out in this morphology.

Fig8. Was this data used in the calibration of the Eq14 and 15 as well?

L303:  bedload axis. Do you refer to a bedload active width?

L313-L317 +Fig9: Did you use Eq14 to calculate the transport rate using the raw apparent velocities, estimating qs for every ensemble of va?
If this is the case, the Eq14 is not calibrated over instantaneous values but over long averages, so It is not statistically sustained. Why don't you use only va?
L325: remove the comma.

Fig10: same as the comment above. The equation is not calibrated for the instantaneous values of the instrument.

L325-328: Do you suggest that the hydrophones are sensitive to the bar formation??

L352: "bedload transport occurs over a very low thickness", please reformulate to: in weak bedload transport conditions the BTMA sampler most likely performed with reduced efficiency … and cite literature.
L365: "during few time" to "little time" or

L378-384: This is very superficial information given in a way to suppress the ADCPs under the hydrophones. And it is only partially true. Please correct according to the general comments 3 and cite accordingly.
L378: (Conevski, et al., 2019)+ (Conevski, et al., (2020, accepted))+ (Conevski, et al., 2020)

L380: (Conevski, et al., 2020)

L385: Instead of "water depth heterogeneity" please use beam footprint heterogeneity

L386: After the filtering (Conevski, et al., 2019) negative values should not exist. But if there is beam heterogeneity this adcp value is rather incorrect and shell be eliminated not used as null. Also if somehow all the beams are sampling the dunes trough (lee side) the measurement is also incorrect if one aims to estimate the bedload transport over a cross-section.

L392: Please order the citations.

L392-395: This is not true. See my other comments. Beside active layer with a height of D50 would assume only rolling of the particles.
L395: You have observed only rolling of particles? Please send one video. I could only imagine this in incipient motion, although It is also impossible. On the contrary, (Lajeunesse, et al., 2010) and (Niño, et al., 1998) report that most of the particles are actually jumping, meaning at least 2D50.

L396: Please do not confuse suspension or suspended load with bedload and its active layer. Saltation is not a suspension.
L397-404: This should be reformulated because it is not true. Please see the general comments regarding the ADCP and the comment above. And cite (Conevski, et al., 2020) for the 3MHz

L408: Indeed, physical samplers sample the dynamical active layer, thus more comparable to the hydrophones and adcps. The usual approach is to integrate the sampled bedload t.rate over the active width and compare to the DTM.

L421: This is very subjective. Conevski et al 2020 report 5min sampling per position using the M9. So in total 30 min, although no bars are presented in the study.

L440: Could you elaborate on how this is not valid for the hydrophones if they are in a fixed position? In the same way, by use of the RTK one could move the ADCPs and estimate the BT velocity, or just subtract the boat velocity from the BT velocity measured by another device or the boat itself. Or if it is certain that one dune has passed below the ADCPs then using the filtering (Conevski, et al., 2019) the data shell be fairly accurate.

L443: water depth not flow depth.

L452-456: Could you please reformulate. Do not really understand how the hydrophones are sensing the bars. I assume you mean change of the dunes which is just changing of the instantaneous transport and the shape of the dune…   It could be another structure, not necessarily a bar… Is this correct?

L459: Please change Conversely to on the contrary.

L462:  This is also partially true. The accuracy is within this range, but this stands for moving boat measurements. The laboratory data suggest lower values until 1mm/s check (Conevski, et al., 2020).

L469-L471: Please correct according to the general comments for the ADCP. If **strongly** correlated wit the grain size, please show the correlation.

**References**

**Banhold Karin [et al.]** Underestimation of sand loads during bed-load measurements—a laboratory examination [Conference]. - 2016. - pp. 1036-1041.

**Brumley B. H. [et al.]** Performance of a broad-band acoustic Doppler current profiler [Journal] // IEEE Journal of Oceanic Engineering. - 10 1991. - Vol. 16. - pp. 402-407. - ISSN: 0364-9059.

**Conevski S. [et al.]** Acoustic sampling effects on bedload quantification using acoustic Doppler current profilers [Journal] // Journal of Hydraulic Research. - 2020. - pp. 1-19.

**Conevski S. [et al.]** Laboratory investigation of the apparent bedload velocity measured by ADCPs [Journal] // Journal of Hydraulic Engineering. - 2019.

**Conevski S. [et al.]** TOWARDS AN EVALUATION OF BEDLOAD TRANSPORT CHARACTERISTICS BY USING DOPPLER AND BACKSCATTER OUTPUTS FROM ADCPS [Journal] // Journal of Hydraulic Research. - (2020, accepted).

**Conevski Slaven [et al.]** Bedload Velocity and Backscattering Strength from Mobile Sediment Bed: A Laboratory Investigation Comparing Bistatic Versus Monostatic Acoustic Configuration [Journal] // WATER. - 2020. - 3318 : Vol. 12.

**Curran Joanna C. and Wilcock Peter R.** Characteristic dimensions of the step-pool bed configuration: An experimental study [Journal] // Water Resources Research. - 2005. - Vol. 41.

**Frings Roy M and Stefan Vollmer** Guidelines for sampling bedload transport with minimum uncertainty [Journal] // Sedimentology. - 2017.

**Gaeuman D. and Jacobson R. B.** Acoustic bed velocity and bed load dynamics in a large sand bed river [Journal] // J Geophys Res Earth Surf. - 2006. - F02005 : Vol. 111(F2).

**Gaweesh Moustafa and C. van Rijn Leo** Bed-Load Sampling in Sand-Bed Rivers [Journal] // Journal of Hydraulic Engineering-asce - J HYDRAUL ENG-ASCE. - 12 1994. - Vol. 120.

**Guerrero M., Lamberti, A.** Flow field and morphology mapping using ADCP and multibeam [Journal] // J. Hydraulic Eng.. - 2011. - 137(12). - pp. 1576-1587.

**Lajeunesse E., Malverti L. and Charru F.** Bed load transport in turbulent flow at the grain scale: Experiments and modeling [Journal] // Journal of Geophysical Research: Earth Surface. - 2010. - 4 : Vol. 115.

**Nin˜o Y and Garcı`a M** Experiments on saltation of sand in water. [Journal] // J Hydr Eng. - 1998. - Vol. 124. - pp. 1014–1025.

**Parker Gary** ID SEDIMENT TRANSPORT MORPHODYNAMICS with applications to RIVERS AND TURBIDITY CURRENTS [Book]. - [s.l.] : St. Anthony Falls Laboratory, Mississippi River at 3rd Avenue SE, 2004.

**R. Lee Gordon RD Instruments** Acoustic Doppler Current Profiler Principles of Operation A Practical Primer [Report]. - San Diego, California : RD Instruments, 1996.

**Rennie C. D., Millar R. G. and Church M. A.** Measurement of bed load velocity using an acoustic doppler current profiler [Journal] // Journal of Hydraulic Engineering. - 2002. - 128 : Vol. 5. - pp. 473-483.

**Rennie C.D and Villard P. V.** Site specificity of bed load measurement using an acoustic Doppler current profiler [Journal] // J. Geophys. Res.. - 2004. - F03003 : Vol. 109.

**Rennie C.D. [et al.]** "Calibration of aDcp apparent bedload velocity to bedload transport rate" in Gravel-Bed Rivers [Book Section] // Processes and Disasters. - [s.l.] : Tsutsumi, D., and Laronne, J., Wiley., 2017.

**Shiel F. Douglas** Aquatic Habitat Bottom Classification Using ADCP [Journal] // JOURNAL OF HYDRAULIC ENGINEERING. - 2010. - pp. 336-342.

**Urick Robert J.** Principles of Underwater Sound [Book]. - [s.l.] : McGraw-Hill, 1983.

**Wilcock Peter R., Kenworthy Stephen T. and Crowe Joanna C.** Experimental study of the transport of mixed sand and gravel [Journal] // Water Resources Research. - 2001. - Vol. 37. - pp. 3349-3358.

---

## Referee Comment (RC3) · Paul Grams (Referee) · 4 Jan 2021

This is a review of the manuscript, "Relevance of acoustic methods to quantify bedload transport and bedform dynamics in a large sandy-gravel bed river" by J. LeGuern and others submitted for publication in Earth Surface Dynamics. The article compares measurements of bedload obtained by four different methods for a study site on the Loire River. The methods are physical bedload samples collected with a large isokinetic sampler, measurements of bed-surface velocity with ADCP, measurements of dune celerity with singlebeam sonar, and measurements of acoustical power by hydrophone. This is a very nice dataset and the comparison presented is interesting and should be

useful to future studies of bedload transport, especially in large rivers. I think the article should be published, with revisions. My major comments are as follows: 1) I think the conclusions regarding dune tracking are more specific to how the method was applied in this study and to this field site. Thus, I don't think that general conclusions about the accuracy or utility of that method can be claimed. 2) The authors state that acoustical power was not affected by water depth, but no data are presented to demonstrate that conclusion. Because acoustical power is expected to be affected by water depth, that should be demonstrated, rather than asserted. 3) Finally, the article needs a careful proof reading for minor grammatical errors. I made some corrections, but did not do a thorough edit.

Please see additional comments in the attached pdf version of the manuscript with annotations.

Paul Grams December 30, 2020

Please also note the supplement to this comment:
https://esurf.copernicus.org/preprints/esurf-2020-77/esurf-2020-77-RC3-supplement.pdf

---

## Author Comment (AC1) · 8 Feb 2021

**General comments (GC)**

*GC1*

*"Specifically, in relation to the hydrophone method, I would like to see a clearer explanation of the method and some data that demonstrate the variability of the power-spectral density between hydrophone recordings."*

We evaluated the variability of the PSD for all hydrophone records (n=448, Figure1a). This variability is illustrated by computing the central frequency and the peak frequency in the frequency band of bedload (15-350 kHz) (Figure 1b). The central frequency is affected by the grain size and vary between 50 and 270 kHz with a median value of 140 kHz. The first quartile and the third quartile are 127 and 154 kHz respectively that reflect a small variability of the central frequency. This is in agreement with the median grain size of sediment ($D_{50}$) distribution that vary between 0.3 and 3 mm with a median value of 0.9 mm (Q1=0.8 mm and Q3=1.1 mm).

[Figure]

Figure 1: a) Power Spectral Density (PSD) for each hydrophone drift of the study (n=448), the median PSD in black; b) Distribution of the frequency peak and central frequency of each PSD.

*GC2*

*"I would like the paper to describe if there are significant differences observed between the frequency spectrum of the studied sandy gravel-bed river and the frequency spectra in the comprehensive literature on gravel-bed rivers. How does the different grain size distribution affect the observed frequency spectrum?"*

Different frequency spectrum from three different rivers are plotted in the Figure 2a. As expected by Thorne (1986), the central frequency decreases with the $D_{50}$ (Figure 2a). We acquired 98 concomitant hydrophone records and sediment samplings and we compared these measured values to the theoretical relation of Thorne (2014) (Figure 2b). For the investigated range of $D_{50}$, the central frequency measured are in line with the theory (82% of the dataset are in the factor 2). For this range of grain size, Geay et al. (2020) found that rivers investigated did not follow Thorne's law because of the attenuation of high frequency sound waves in these steep rivers. The shift of the bedload frequency band towards high frequencies (figure 2a) means more bedload signal in this area of the PSD. However, the slope of the study reach is mild and the turbulence is relatively low. Therefore, the high frequency sound wave are not attenuated and the acoustic power in high

frequency region is not neglected. The use of hydrophone method in large lowland sandy-gravel bed rivers could be an asset considering the bedload frequency band of sands.

[Figure]

Figure 2: a) Comparison of PSD from 3 rivers with varying $D_{50}$; b) Comparison between measured central frequency of PSD of the Loire River and related $D_{50}$ with relation of Thorne (1986).

We propose to add the figure 2a to the section 4.1. in order to provide PSD information related to other rivers, and also a paragraph: "*Moreover, the median PSD differ from the Isère River (Petrut et al., 2018) and from Drau River (Geay et al., 2017). These rivers are characterised by coarser sediments (see Fig. 5a) and the central frequency of the PSD are decreasing for an increasing $D_{50}$. These observations are in line with Thorne's (1986) theory. The central frequency of the median spectrum of the Loire is about 140 kHz. The frequency band of the bedload is shifted towards high frequencies due to finer grain size.*"

**Specific comments (SC)**

*SC1*

*Lines 57-58 "to estimate the capacity of acoustic signals to detect the bedload axes on relatively wide cross-sections for various discharge conditions".*

*"Are you referring to estimating the accuracy of acoustic methods for measuring bedload transport rates for wide transects across a range of flow conditions?"*

Yes, we are referring to the accuracy of acoustic methods to measure cross sectional variations of bedload fluxes over various discharge conditions. This sentence was modified in the revised version of the paper.

*"to estimate the accuracy of acoustic methods to measure cross sectional variations of bedload fluxes for various discharge conditions;"*

*SC2*

*Line 116 "giving better results"*

*"How do you know that they are better results? Please provide a clearer description of the method used to determine data quality."*

We just wanted to refer to Jamieson et al. (2011, p. 1066) observations here: *"A positive bias in bed velocity magnitude could be generated by interpolating bed velocity magnitude alone (i.e., without direction), where inconsistent bed velocity directions (which are typically higher in regions of zero to low bed velocity, where a moving bed is more difficult to detect with the ADCP) generates higher than expected bed velocity.*"

We changed "*giving better results*" by "*limiting the over estimation*".

*SC3*

*Lines 147-148 "and could allow a better understanding of the apparent bedload velocity gradient along bedforms."*

*"It's not clear to me what this means. Why would this give a better understanding of the bedload velocity gradient along bedforms? Is it because the ADCP footprint is relatively small compared with the bedform dimensions? If so, please state typical bedform dimensions relative to the footprint."*

This is exactly what we mean. We propose to change this sentence by: "*This configuration permitted to fix the footprint for each beam to about 0.0046 m² and a distance 146 of 0.56 m between opposed beams. This allowed to describe apparent bedload velocity with a finer accuracy especially in the presence of bedforms of 0.2 m height and 3.9 m long (in average)."*

SC4

*Lines 161-162 "The determination of a proxy to evaluate sediment transport directly from DTM measurements is difficult."*

*"Explain why it was difficult."*

Because there are several dune parameters such as dune celerity, height or length that influence bedload rate and all these parameters are evolving according to discharge variations. We propose to modify this sentence by: "*The determination of a proxy to evaluate sediment transport directly from DTM measurements is difficult because dune migration is function of several parameters. A semi-empirical equation that integer these parameters was used to compare bedload transport rates with the reference measurement.*"

SC5

*Lines 240-241 "and the integration of median PSD over a wider range of frequency in the present study."*

*"What is meant by median? I'm not sure what this is referring to and I think it needs to be explained more clearly. Is the PSD integrated across the median value of each frequency bin?"*

To make things more clearly, we decided to add this comment to the final version: "*For each drift, a spectral probability density is computed (Merchant et al., 2013). Then a median Power Spectral Density is determined as done in Geay et al. (2017). Median PSD are preferred to mean PSD as it enables to filter rare and powerful acoustic events such as the hydrophone impinging the riverbed.*"

SC6

*Lines 251-257 "The comparison can be made between indirect methods to discuss the acceptability of the BTMA reference. The apparent bedload velocity and the acoustic power are not well-correlated with mean dune morphological parameters (dune celerity and height). The aDcp method is measuring the apparent velocity of the grain being transported from the stoss to the lee side of a dune. It must be noted that apparent bedload velocity is higher than dune celerity with about a factor 100, whereas the grain size (D50) is smaller than dune height with the same order. Therefore, sediments that are 100 times smaller than dune height allows the dune migration with a celerity 100 times smaller than their own celerity."*

*"I'm not sure this section is necessary and I struggle to understand some of the arguments being made. What is meant by the acoustic power and apparent bedload velocity not being well correlated with the morphological parameters? Where is this shown in the data? The comparison between apparent bedload velocity and dune celerity is interesting but I think most readers would already understand that the bedload velocity is much smaller."*

We wanted just to mention that relations between dune parameters (height and celerity) and acoustic methods are not consistent. We chose not to show these relations with figures to avoid overload of the paper. This can be illustrated by a short table of the coefficient of determination (COD) for each relation. We added this table.

Table 1: Coefficient of determination between dune parameters and acoustic methods (log values).

|  | P | $V_a$ | $q_{s\ BTMA}$ | $H_{dune}$ | $C_{dune}$ |
|---|---|---|---|---|---|
| P | - | - | - | - | - |
| $V_a$ | 0.76 | - | - | - | - |
| $q_{s\ BTMA}$ | 0.70 | 0.48 | - | - | - |
| $H_{dune}$ | 0.20 | 0.27 | 0.16 | - | - |
| $C_{dune}$ | 0.22 | 0.24 | 0.36 | 0.22 | - |

We agree that the following sentence is confusing and not necessary. We wanted to show differences between the magnitude order of dune celerity and apparent bedload velocity, keeping in mind that the apparent bedload velocity participate to dune migration. Just to be clear, you mentioned that bedload velocity is smaller but this is the opposite, the bedload velocity is higher than dune celerity. We modified the text, keeping just this part of the sentence: "*It must be noted that apparent bedload velocity is higher than dune celerity with about a factor 100.*"

*SC7*

*Lines 257-258: "On the other hand, the apparent bedload velocity is positively correlated with the acoustic power. The RMA regression model explains 76% of the dataset dispersion (Fig. 6a)."*

*Line 261 "and water discharge explain 71% of the dataset dispersion"*

*"What it meant by 'explains 76% of the dataset'? This needs to be accurately described. This paragraph needs to be re-written."*

The COD (R²) of the RMA regression is equal to 0.76 this mean that the equation of the RMA regression (model) determines 76% of point distribution (scattering). We modified these two sentences by: *"The COD of the RMA regression is equal to 0.76.*"; and "*The COD of the RMA regression established between BTMA bedload rates and water discharge is 0.71.*"

*SC8*

*Line 286 "there was no reference measurements"*

*"Why not? Please explain. Can you please add labels, e.g. S1, to the BTMA data points in both panels of Fig. 8 to clearly identify the reference measurements? Why is there no S2 in Fig. 8a? Why are reference measurements missing from Fig. 8b?"*

As mentioned in the material and method section, BTMA method is very time-consumming and it is not possible to measure all sampling points for each field survey. This is the case for S2, S5 and S6 on the figure 8a and also for S6 on the figure 8b. We added sampling points on figures 8a et 8b as requested:

[Figure]

*Line 294 I would like to see some discussion as to why the hydrophone method is producing larger values of unit bedload rate compared with the BTMA measurements. Just a thought, but could it be related to the omnidirectional hydrophone picking up higher noise magnitudes that are not directly below the boat?*

A hydrophone senses every noises that are propagating in the water column. Therefore, the hydrophone can sense noises that are far away from its location. Noises are more and more attenuated with increasing distance (Geay et al., 2019). Particularly, when there is few bedload noises close to the hydrophone, the hydrophone can sense bedload noises that are generated far away. This behaviour could explain why the hydrophone tends to overestimate bedload fluxes when bedload fluxes are weak.

*Lines 303-305 "For this survey, acoustic signals (i.e. acoustic power, apparent bedload velocity) followed the same evolution pattern as isokinetic samplers along the cross section except for S3."*

*"Again, why is this the case? Is it due to a lack of directionality?"*

As the aDcp method, isokinetic sampler method is a "point/local" measurement which is function of the local grain size. The sediment grain size variability is important to explain the spatio-temporal variability in bedload sediment dynamics in large sandy gravel bed rivers. For instance, we made some measurements where one BTMA caught some sediment transported and the other one (located about 4 m away on the same cross section) was empty due to coarser grain size at this specific location. So, it is clear that an uncertainty is due to the limited spatial resolution of these two methods compared with hydrophone method that integer longitudinal variability of bedload (drifts are about 30 m long depending of flow velocity). Moreover, the presence of bedforms, specifically bars, increases the bedload (and grain size) variability (Venditti et al., 2012; Cordier et al., 2020).

_"Line 325 Immediately downstream of the bar there are bedload transport values that are higher than those observed further downstream. Is this due to the omnidirectional hydrophone picking up bedload noise from the bar upstream?"_

It is possible that the hydrophone senses the noise that are generated at the bar top and that acoustic power decreases smoothly when increasing the distance from the bar edge. The small increase of noise could be the noise of a dune that fall on the avalanche face (lee face) of the bar when the hydrophone was over the bar front, but we can only make assumptions here.

All other technical corrections mentioned by the referee were adopted in the final version of the paper.

**References**

Cordier, F., Tassi, P., Claude, N., Crosato. A., Rodrigues, S., and Pham Van Bang, D.: Bar pattern and sediment sorting in a channel contraction/expansion area: Application to the Loire River at Bréhémont (France). Advances in Water Resources, 140, https://doi.org/10.1016/j.advwatres.2020.103580, 2020.

Geay, T., Belleudy, P., Gervaise, C., Habersack, H., Aigner, J., Kreisler, A., Seitz, H., and Laronne, J. B.: Passive acoustic monitoring of bed load discharge in a large gravel bed river, J. Geophys. Res.: Earth Surf., 122 (2), https://doi.org/10.1002/2016JF004112, 2017.

Geay, T., Michel, L., Zanker, S., and Rigby, J. R.: Acoustic wave propagation in rivers: an experimental study. Earth Surface Dynamics, 7 (2), 537–548, https://doi.org/10.5194/esurf-7-537-2019, 2019.

Geay, T., Zanker, S., Misset, C., and Recking, A.: Passive Acoustic Measurement of Bedload Transport: Toward a Global Calibration Curve?: J. Geophys. Res.: Earth Surf., 125 (8), https://doi.org/10.1029/2019JF005242, 2020.

Merchant, N. D., Barton, T. R., Thompson, P. M., Pirotta, E., Dakin, D. T., and Dorocicz, J.: Spectral probability density as a tool for ambient noise analysis. The Journal of the Acoustical Society of America, 133, https://doi.org/10.1121/1.4794934, 2013.

Petrut, T., Geay, T., Gervaise, C., Belleudy, P., and Zanker, S.: Passive acoustic measurement of bedload grain size distribution using self-generated noise. Hydrol. Earth Syst. Sci., 22, 767-787, https://doi.org/10.5194/hess-22-767-2018, 2018.

Thorne, P. D.: Laboratory and marine measurements on the acoustic detection of sediment transport, J. Acoust. Soc. Am., 80(3), 899, https://doi.org/10.1121/1.393913, 1986.

Thorne, P. D.: An overview of underwater sound generated by interparticle collisions and its application to the measurements of coarse sediment bedload transport, Earth Surf. Dyn., 2 (2), 531–543, https://doi.org/10.5194/esurf-2-531-2014, 2014.

Venditti, J. G., Nelson, P. A., Minear, J. T., Wooster, J., and Dietrich, W. E.: Alternate bar response to sediment supply termination. Journal of Geophisical Research, 117, F02039, https://doi:10.1029/2011JF002254, 2012.

---

## Author Comment (AC2) · 8 Feb 2021

**General comments (GC)**

GC1

a)  *"In the text, there is no info about the standard deviation of the particle's sizes of the bedload material. I would also like to see the range of the particle Reynolds number (Rep), the shear velocity or stress and the critical shear velocity or stress. It could be also interesting if the authors give information about the reference shear stress at the study site (Parker et. al 1982). Note that this is similar to the critical shear stress but not the same."*

As expected by the referee, we added some details about sediment characteristics in the text.

"*It varies between 0.3 and 3.1 mm with a standard deviation of 0.4 mm. The $D_{90}$ is much variable with a median value of 3.3 mm ranging from 0.5 to 15.7 mm. Hydraulic conditions at the sampling points varied with discharge; the median water depth and flow velocity are 1.6 m and 0.9 m.s$^{-1}$, respectively.*"

In order to limit the number of figures in the paper, we proposed to add the figures 1a and 1b (please see below) in the present author's response. Colleagues interested will find all the details and figures in the present document available on the website.

Hydraulic conditions and sediment characteristics allowed to compute the Shields stress ($\tau^*$) and its critical value ($\tau_c^*$) for 312 samples with the following equations (figure 1b):

$$\tau^* = \frac{D.S}{RD_{50}};$$

Where D is the water depth, S is the slope and R is the submerged specific gravity of the sediment (1.65).

$$\tau_c^* = 0.22\, Re_p^{-0.6} + 0.06\, e^{(-17.77 Re_p^{-0.6})};$$

Where $Re_{p}$ is the particle Reynolds number:

$$Re_p = \frac{\sqrt{RgD_{50}}\, D_{50}}{\nu};$$

With *g*, the gravitational acceleration (9.81 m.s$^{-2}$) and $\nu$ the kinematic viscosity of water (10$^{-6}$ m².s$^{-1}$). The Shields stress related to the $D_{50}$ of these 312 samples was higher than the critical Shields stress for all hydraulic conditions measured (figure 1b). The sediment (rather fine) is characterized by a significant mobility for various discharge and flow velocity conditions.

[Figure]

Figure 1: a) Distribution of grain size statistical parameters for n=447 samples. b) Shields parameter as a function of particle Reynolds number ($D_{50}$) with Shields curve of critical parameter.

Table 1: Distribution of characteristics sediment diameters for 447 samples and hydraulic conditions (water velocity V and water depth D) at sampling points.

|  | $D_{10}$ (mm) | $D_{50}$ (mm) | $D_{90}$ (mm) | V (m.s$^{-1}$) | D (m) | $R_{ep}$ |
|---|---|---|---|---|---|---|
| Median | 0.4 | 0.9 | 3.3 | 0.9 | 1.6 | 111.3 |
| Minimum | 0.2 | 0.3 | 0.5 | 0.2 | 0.4 | 25.1 |
| Maximum | 0.9 | 3.1 | 15.7 | 1.4 | 4.8 | 705.4 |
| Standard deviation | 0.1 | 0.4 | 2.3 | 0.2 | 0.9 | 86.0 |

b)      *"There are quite a few comments about the measurement capability of the instruments related to the different PSD of the bedload particles, but not details about the PSD of the sampled bedload surface."*

We presume that you refer to Particle Size Distribution/ Grain Size Distribution by mentioning PSD (and not to Power Spectral Density as in the paper). We are not sure to understand your comment. The GSD (grain size distribution) is mentioned in the *GC1a* point you raised.

Anyway, we did not investigate in this paper the specific potential link between GSD and the capability of the instruments. As a first approximation we've seen on some hydrophone records that for a given PSD the shape of the frequency distribution curve is different according to the GSD. This is a very interesting point we will investigate in another contribution.

c)      "I have another question about the boat/floating structure given on Fig2b: How was it fixed and how intensive was the drifting that you mention several times (in terms of meters)?"

The figure 2b of the paper illustrated the boat used for BTMA and aDcp sampling. As mentioned line 89, this 20 m long boat is fixed with 2 anchors, one in the front of the boat and 1 anchor at the back. The aDcp and its floating structure were fixed and tightened to the boat with a rope. There was no drifting of this structure during measurements (only small lateral movements of the boat were possible). We mentioned drifting only in the hydrophone protocol. Passive acoustic measurements were acquired with a smaller boat, turning off the engine and drifting at water velocity during more or less 30 seconds (about 50 m).

a)      "I was wondering how this calibration coefficient (α=2) was adapted. Does it take in consideration all the factors numbers above (check Banhold, et al., [2016] for detailed analysis of a very similar sampler in Germany)? It would be good to see a sentence regarding this issue. Calib coefficient of means that the samples underestimate 100 % of the real sediment transport."

b)      and d) "Based on which tests? I could not find the citation online…Boiten 2003? It looks like lecture notes, so inside it must be stated how these coefficients are defined and related to another study. And if some experiments exist, were those conducted in similar conditions as your study site?"

Boiten (2003) is a textbook describing lecture notes of Delf University (https://www.taylorfrancis.com/books/hydrometry-boiten/10.1201/9780203971093). We added the DOI in the reference list. Details of the calibration procedure are in two Dutch reports that I cannot find (Delft Hydraulics Laboratory, 1958 and 1969). But, some outlines are described in de Vries (1979). The first calibration tests consisted to compare the BTMA bedload catches with the average flume sediment transport. In a second time, tests were made by weighing catches in a sediment trap of the same size than sampler mouth. Sediment mixtures used during these tests was coarser than bedload of the Loire River with D50 varying between 2.5 mm and 6.4 mm. Both test series concluded to the same calibration coefficient of 2 (efficiency of 50%). This means that actual transport is two times higher than those measured with the sampler. The average efficiency of a basket sampler as the BTMA sampler is about 45% (Hubbell, 1964). These tests established a calibration curve linking unit bedload transport rates with caught volumes of sediments. It is mentioned in Boiten (2003) that this calibration factor was not including the possible losses of sediments finer than 0.3 mm (size of the mesh). The sand loss was estimated during flume experiments with a similar samples by Banhold et al. (2016) to 50% in average but with a mesh size of 1.4 mm and varied sediment mixture (D50=[0.8-10] mm). There is no detail about the hydraulic coefficient of the BTMA which is the ratio between velocity in the sampler and flow velocity in de Vries (1979). But, author argued that the BTMA construction promote the transport coefficient at the expense of hydraulic coefficient. In comparison, the Helley-Smith sampler tend to overestimate bedload with the ratio between the flume sediment transport and the sampler sediment transport that vary with water velocity from 1.2 to 2.6 (Helley and Smith, 1971). The BTMA was already tested on the Loire River (Gautier et al., 2008 ; Claude et al., 2012) and its results were compared with others estimation of bedload sediment rates.

The sampling error was estimated to 30% with 10 observations, 20% with 20 observations and 9% with 100 observations (Eijkelkamp, 2003). This stochastic uncertainty can be evaluated for each sample point with the standard deviation of unit bedload ($\sigma_{q_{s\,BTMA}}$) and the number of bedload samples ($n$) (Frings and Vollmer, 2017):

$$E_{q_{s\,BTMA}} = \sigma_{q_{s\,BTMA}} \; n^{-0.5} \; ;$$

The mean calculated value of stochastic uncertainty is about 30% of the mean bedload rates measured. It varies between 12% and 100% and increases when bedload transport decreases. The uncertainty given by the constructor is not far away from mean calculated value. To conclude, we are not able to describe with details how the calibration factor is fixed but we know that this calibration factor don't take into account the under-estimation of fine sands (less than 0.3 mm).

This sentence was added to the final version:

"*Suggested values of α and b were adopted from Boiten (2003) which mentioned that the trap efficiency factor not include the possible losses of sediment finer than 0.3 mm (mesh size opening). This calibration factor come from successive calibration procedures concluding to the same calibration coefficient of 2 (de Vries, 1979).*"

"*The uncertainty of unit bedload was calculated using the equation of the stochastic uncertainty from Frings and Vollmer (2017).*"

c)      "Is there a defined procedure for sampling? How many samples are sufficient to minimize the uncertainty (Frings, et al., 2017)?"

The defined procedure for sampling is described in Boiten (2003). We used 2 BTMAs to decrease the sampling time on each sampling points. At each sampling point 10 samples were collected with each BTMA (so 20 in total) and volumes of each samples were measured *in-situ* with a graduated cone (Imhoff cone). All samples volumes from each BTMA were merged for sieving analysis (leading to 2 sediment samples per sampling point; one for each BTMA). Then, the average volume of caught sediments from the 2 BTMAs was computed and converted into instantaneous unit bedload rates using equation 1. This equation is extracted from the calibration curve mentioned above and adapted to convert volume into mass of sediment.

We added some details of the procedure in the final version of the paper: "*At each sampling point, 10 samples were collected with each BTMA (so 20 in total) and volumes of each samples were measured in-situ with a graduated cone (Imhoff cone). Collected volumes were integrated over at least 2 minutes. All samples volumes from each BTMA were merged for sieving analysis (leading to 2 sediment samples per sampling point; one for each BTMA). Then, the average volume of caught sediments from the 2 BTMAs was computed and converted into instantaneous unit bedload rates using Eq. (1):*"

e)      "What are the conditions when the sampler would malfunction (e.g. weak transport conditions or too abundant)? There are many examples of low transport rates could be underestimated or overestimated by the physical samplers. Although most of the literature suggests that in weak transport conditions the samplers would over-estimate (e.g., due to digging in the bed), sometimes in sandy rivers, we have noticed many samples to under-sample the total transport (observing videos and total mass caught)."

We fully agree with your comment that in weak transport conditions the samplers tend to under-estimate bedload. Video records showed a very thin bedload layer for these conditions and the mouth of the sampler is not always well posed on the river bed, letting through particles underneath the mouth sampler. This occurs when sediments are coarser, when dunes are present and also immediately downstream a bar front where the riverbed (lee effect).During flood event, it is difficult to see the mouth's sampler on video records and adding a light was not a solution because there are lot of particles in saltation/suspension near the bed for these hydraulic conditions.

f)      g) "Given the Eq. 1 I was wondering if you have done PSD and weight/sieving analysis in the laboratory for each sample? If so, why not using only the dry mass? Besides the sieving, there is no information on how you have measured the other parameters, such as the volume V."

As mentioned in the response *GC2b*, we did sieving analysis on the total volume of caught sediments for each BTMA of each sampling point (2 grain size analyses per sampling point). Sampling volumes were merged (one sample for BTMA1, another for BTMA2 for each sampling point) so we are not able to use dry mass in laboratory. Bedload volumes are directly measured in the field with a graded cone.

h)      "In the appendices, you give information on how many samples are averaged in the given value. But there is no explanation of how the samples from the two BTMA are used in the processing. Please give information about that."

Please see our previous comment (*GC2b).*"

**GC3**

a)      "The post-processing procedure only partially follows some of the latest findings given in (Conevski, et al., 2019) and (Conevski, et al., 2020). Although the authors tried to implement some of the procedure there are some miss-steps. The de-spiking and the filtering procedure (discarding the raw BT velocities that are in the opposite direction of the flow) given in these two studies improved the correlation with the samples and gave more realistic velocities. It seems that the filtering procedure automatically excluded the negative velocities that occur in the so-called recirculation zone of the dunes (which you also mention). In addition, there is some sort of final filtering that involves analysis of the difference of the depth values registered by each beam, which helped to discard some samples that suffered from beam inhomogeneity (Conevski, et al., 2020)."

We are grateful for this very good idea and also for the reference. We agree that the filtering procedure mentioned in Conevski et al. (2020) would probably improve the results. Nevertheless, we did not have time to develop the code that could be used to process the dataset (initial submission of our paper: 21 of September 2020). We plan to test the filtering procedure detailed in Conveski et al. (2020) in a future paper.

b)      "The projection to the water direction (Wdir GPS) is not clear to me. How these two are defined do you use compass or GPS heading for Wdir and compass heading for Va? Why is this necessary? You say "To avoid compass and GPS issues, and to eliminate the effect of residual lateral displacement", but then you use again: Wdir GPS (GPS ref coordinate system)-bdir BT (compass heading EH). By doing this some of the issues given above are inherited again in the velocity. Also, it is not true that the bedload velocity should align with the water velocity especially in presence of bedforms. If one would like to project one velocity vector on another both should have then the same reference coordinate system or to contain information (rotation and /or translation) about the different reference coordinate system. As far as I understand you use GPS-ENU and Compass – ENU. Note that you are in a fixed position and you do not need water velocity referenced to the BT. Another thing is that if you project over the water velocity the real direction of the bedload is lost, and the final value decreased. Please correct me if I understood wrong."

You are right, there is a mistake here. We corrected $W_{dir\ GPS}$ by $W_{dir\ BT}$ in the equation and the results (figure 3a, plot and RMA equation). So all direction information come from the compass

heading. We agree that by doing this, we reduced the final bedload velocity magnitude. But, the idea was to get the flux velocity and not the bedload velocity magnitude. We are aware that this is an assumption to consider bedload in the same direction of the main flow direction mainly in presence of bar. Again, maybe the filtering procedure mentioned above would lead to better results than this method.

We added: "*This method involve the assumption that bedload is in the same direction than the main flow direction.*"

c)      "The concept of volume and surface scattering seems not to be considered in conceptualizing the discussions. Please check (Urick, 1983), also (Conevski, et al., 2019), characteristics (Conevski, et al., 2020, accepted). Note that the scattering of the surface formed by the immobile particles below the active layer has large impedance and therefore the echo coming from this scattering is dominant. The roughness of this surface could induce false velocity estimation. This error depends on the acoustic parameters of the instrument and the signal processing algorithm. Please have in mind that this is a complex two-phase (surface +volume) scattering process occurring between two scattering regions (Rayleigh + geometric), so it cannot be easily described."

d)      "In the same manner, the concept of internal processing and signal processing is not well addressed. Note that the RDI uses broadband codded pulse technology while M9 uses the pulse-coherent method with some internal modifications such as the use of HD smart pulse. Note that the RDIs BB is able to use much finer resolution systems (Brumley, et al., 1991) in the detection of the bottom signal, thus it should be able to define the better immobile bed surface. Note that the main purpose of the bottom tracking signal is to identify the riverbed and it is developed in that way (R. Lee Gordon, 1996). This is one of the reasons why RDI continuously reports lower velocities than M9, regardless of the frequency (Conevski, et al., 2020). This is also related to the acoustic sampling effect described in the same paper."

e)      "Although it is well known that the grain size is correlated with the backscattering strength even with the bedload particles (Conevski, et al., (2020, accepted); Shiel, 2010), the laboratory investigation (Conevski, et al., 2019; Conevski, et al., (2020, accepted); Conevski, et al., 2020) demonstrated that it is not the only reason for different Doppler / apparent bedload velocities especially in well-developed transport conditions."

We are grateful to all your above comments that would improve the discussion of the paper. We decided to add a sentence in the manuscript based on your comments (GC3 c, d, e):

"*The response of aDcp to bedload transport depends on several parameters The variation of the impulse frequency, the pulse length, beam focusing or associated internal signal processing (Broadband or Narrowband) can lead to different estimation of the apparent bedload velocity for the same sediment transport conditions (Conevski et al., 2020a). These parameters vary from a device to another (RDI/Sontek; Conevski et al., 2020b). As the aDcp pulse sample a volume of the riverbed (Rennie et al., 2002) which can lead to a biased estimation of $V_a$: 1) an underestimation in case of large roughness of the riverbed with most of the reflected pulse is scattered by the immobile particles below the active layer (Conevski et al., 2019); 2) an overestimation in case of high concentration of the bedload layer (Rennie et al., 2017) or sand particles became in suspension near to the riverbed (water bias, Rennie et Millar, 2004). Even if a general trend seems to be highlighted by the river comparison (figure 3a) with an increasing*

*bedload rate as grain size increase for a constant $V_a$, the relationship between grain size and $V_a$ cannot be easily described in response to all variables mentioned above. One explanation of this trend could be that suspended sands could contribute to the bottom tracking signal without being caught by the sampler (Rennie et al., 2017).*"

f) "In the results section, there is no distinction in the Figures between the data measured by the RDI 1200 kHz and the M9 (also which frequency). In Latosinski et .al. (2017) and Conevski et al. (2020) it is clearly indicated that different ADCPs give different mean apparent bedload velocities, and it is not strictly dependent on frequency, but also to the acoustic sampling and resolution, pulse length, cell-profiling, signal processing etc. The laboratory tests are given in Conevski et al. (2019) and Conevski, et al. (2020) also confirm and elaborate on these issues."

The scope of this paper is not to study the impact of frequency or pulse length on the apparent velocity. In that way, we did not mention these details in the figure 3a. But you can find details in the appendix B. About 55% of the dataset was measured with the RDI Rio Grande (1.2 kHz). You can find below the figure 3a adapted with the reference of the used device. It is clear that if we had been able to use the same device during all measurements it would be better.

[Figure]

Figure 2: Adaptation of figure 3a of the original paper with point color in function of aDcp device (Rio Grande/M9).

g) "The timing of 3-10 minutes is enough to get a stable apparent velocity at a single position (Conevski et al., 2019; Rennie, et al., 2002)."

When aDcp measurement is associated to BTMA sampling, the aDcp records during all BTMA sampling time. So, the timing mentioned in your comment is easily reached. Nevertheless, Rennie et al (2002) mentioned that the sampling duration have to be about 25 min to get a reliable estimation of apparent bedload velocity. Whereas Conevski et al. (2019) argued that 3-4 min was sufficient if no bedforms were present. We added a sentence in the manuscript as well as the references:

"*The sampling time needed to get a stable apparent velocity is not well defined and ranging between 3 min in case without bedforms (Conevski et al., 2019) and 25 min (Rennie et al., 2002). In our study the sampling time ranged between 5 and 190 minutes.*"

h) "What do you mean by site-specific? This is partially true, and it is only dependent on the shape of the bedforms and the riverbed surface. Some of these elements could be identified and eliminated by filtering procedure, beam inhomogeneity analysis, surveying several cross-sections before positioning. Conevski et. al (2020) reported a relatively high correlation coefficient with data collected at several different positions and two different rivers. As given in the comments above the ADCPs are more instrument dependent than site-specific."

We agree that filtering procedure could eliminate beam heterogeneity due to bedforms. Here mean that the aDcp calibration differ from a river to another. Even if Conevski et al. (2020) have found high correlation coefficient between bedload rate and apparent bedload velocity for two rivers (Elbe River and Oder River), these two rivers are very similar in term of grain size characteristics, shear velocity, water depth or slope. As mentioned L. 381, there is no general calibration equation linking bedload rates and apparent bedload velocity. Every river have its own calibration curve and the strong dependence of $V_a$ on bedload grain size lead to complicate calibration procedure in heterogeneous sediments (Rennie et al., 2017). We discarded this sentence in the discussion and added comment: "*Recent works have been carried out on two rivers (Elbe, Oder) very similar to the Loire River in term of grain size characteristics, flow and shear velocity, and water depth (Conevski et al., 2020a). Even if the correlation between apparent bedload velocity and bedload rates is significant, this calibration equation was obtained from two very similar rivers. Despite these observations, there is no general agreement between bedload rates and apparent velocity (Rennie and Villard, 2004; Rennie et al., 2017).*"

i) "The water bias could be assessed by checking the raw water velocity and backscattering echo values from the last available cell, together with the camera data. Also one should expect rather high suspended sediment concentration in the entire profile (above 500 mg/l) to have the effect of water bias (Rennie, et al., 2002; Gaeuman, et al., 2006). Although this effect has not been investigated in laboratory conditions it is expected that the long BT pulse contains more energy and therefore the signal penetrates this suspended layer. Once it penetrates the scattering reflected both, from the active bedload layer (which has an absolutely higher concentration than the SSC right above) and the surface scattering of the immobile particles below, is absolutely stronger and easier to detect in the autocorrelation as part of Doppler estimation methodology. What was the SSC concentration in some of the cases? Could you upload a video of the sampling area?"

It is a very interesting comment but it is out beyond our competence domain. We believe that these investigations should be carried out experimentally (in a flume study). We are not able to provide an estimation of SSC because the paper mainly focuses on bedload.

*j) The kinematic model.*

*Shear velocity.*

*The semi-empirical approach analytically derived by van Rijn is fairly valid as long as the roughness is assumed to be 3D90. Since you have the ADCP right above the dune a log-law fitting might effectively correct or the Reynolds method could be also applied (Guerrero, 2011). By obtaining the velocity profile right above the dune stoss side, the drag effect from the bedforms might be negligible, thus the shear stress would be only related to the motion of the grain. Not sure if this is feasible in your case, but it is an idea.*

This is a good idea but we think that it is difficult to perform these measurements in field conditions. In our case, the positioning of the aDcp precisely over dune stoss side is difficult because dunes are rather small. Moreover, the sampling points are defined before going on the field (GPS coordinates) to be sure to compare diachronically the same sampling points (to analyze the evolution of bedload for various discharge conditions). We understand that using van Rijn's approach implies that roughness is assumed to be $3D_{90}$. We could consider the effective roughness defined by the same author (van Rijn, 1984b) as the sum of the grain roughness and the form roughness in order to compute the Chézy coefficient and then the shear velocity related to form roughness. Unfortunately, we don't have bedform geometry for each BTMA/aDcp comparison. This makes impossible this computation but remains a very nice field of investigation.

*It seems that the formulas you use for the critical shear velocity do not involve the hiding effect function. Note that you have sand and gravel in the bed mixtures, and therefore a fractional transport. This might not be crucial for only determining the active layer thickness but it is worth checking because the active layer thickness might increase due to the decrease of the critical shear velocity of the gravel particles, in a presence of sand (Wilcock, et al., 2001; Curran, et al., 2005).*

As you mentioned, the shear velocity here is computed to assess the transport stage parameter (Van Rijn, 1984) and then the active layer thickness. Taking into account the hiding effect could be a good idea to see the influence of the composition of the riverbed on the active layer thickness, specifically in reaches characterized by coarse and poorly sorted sediments. Actually, we think that is not crucial in this paper because 1) the proportion of gravels in our sediment mixtures remains not sufficient to induce a hiding effect (16% of small gravels and 84% of sand in average) and 2) The size of the gravels remains quite fine (median value of $D_{90}=3.3$ mm).

*Porosity.*

*Which value do you use for the porosity of bedload concentration in the Eq. 4? A constant bedload concentration of 0.65 is utterly false. Frozen-core bed surface or substrate samples were taken from riverbed have a porosity of 0.4, which simply means the concentration of 0.6. In the same context perfectly, packed particles without consistent shape hardly reach porosity less than 0.3. In the kinematic model, the concentration value is the value of the concentration in the instantaneous (dynamical) bedload active layer. This value shall be at a maximum lower than 0.6; Van Rijn (1984) reports this as maximum bedload concentration. Rennie and Villard (2004) reported 0.1-0.15 in their model. Note that this is not true for the porosity of the dunes; it is also stated in the paper that you make a distinction between the "dynamical active layer" (I am calling it instantaneous, or max saltation height) and the event -scale exchange layer (e.g., this includes the height of the dunes)*

We used the constant value of bedload concentration (0.65), defined by van Rijn (1984) as a maximum. Of course, it varies according to hydraulic conditions. It also differs between bedforms (e.g. dunes and ripples; Simons et al., 1965). Here, our goal was not to determine the exact concentration or the thickness of the active layer precisely but to compare methods. So, we simplified the equation by considering concentration of the bed as a constant. However, we

decided to take into full consideration your suggestions relative to the bedload concentration and apparent velocity. Please see SC8 for full details.

**GC4**

a)      "The river morphology and the presence of bars (that actively migrate) suggest the dune formation throughout this cross-section would be 3-dimensional, which means that I would not expect straight line dune wavefront. Is it correct?"

Partially. Actually, the shape and angle of the dune front will mainly depend on flow depth, velocity and turbulence (and obviously the Froude number). The 3D dunes (sinuous crest) will develop mainly at high flow velocities with reduced water depth. The flow velocities on the site are relatively low (0.8 m.s$^{-1}$ in average) due to the slope and width of the channel investigated. So, most of the bedforms present on the site are characterized by a relatively straight crest (2D dunes defined by Ashley, 1990). Nevertheless, 3D dunes can be rarely present close to obstacles for instance.

b)      "On the other hand, the DTM method that you use is built based on well-developed 2D dunes. Could you please elaborate on this issue? I am afraid that maybe only the dune measurements in the thalweg zone would be most reasonable to use in the comparison with the other two techniques?"

We understand your comment. On the Loire River the DTM method was used on several sites because the 2D dunes were recognized on all these sites (multibeam echosoundings). Please see the following publications: Claude et al., 2012 & 2014; Rodrigues et al., 2012 & 2015; Wintenberger et al., 2015. If dunes are present in the thalweg for all discharge conditions, they develop rapidly on the back of bars since these latter are submerged (Le Guern et al., 2019). That means that most of the riverbed is covered by dunes when bars are submerged. Although the significant width of the channel makes difficult to perform multibeam echosoudings, we did a couple of them, in 2018, on some longitudinal tracks located on the back of bars present in the study site (please see figure below). As it can be seen on this figure, the dunes are clearly 2D dunes for the 3 longitudinal tracks presented.

[Figure]

Figure 3: View of a multibeam longitudinal track measured on the study site (scale=about 300 m long to 4 m width).

c)      "Why not compare the bedload transport integrated over the entire cross-section from the hydrophones and ADCPs vs the dune tracking of only several longitudinal profiles (e.g., close to the thalweg)."

We guess that the elements given previously (see GC4b) show that this would probably lead to a bad estimation because the dunes present close to the talweg will have different morphological features (height, length) and celerity than the dunes present on other parts of the riverbed (stoss side of bars for example).

**GC5**

a)      "What is the advantage of drifting of the hydrophones? Isn't there less ambient noise (not bedload flux noise) if they stay fixed? Ok, I read in the Geay et al. (2020) that it is actually the opposite... I did not find a comparison of both cases. How was drifting performed and why the waves hitting the boat or produced by the boat did not affect the measurements?"

By moving at the water velocity, the ambient noise is reduced in comparison with a fixed position. The drift is performed as follow:

-       positioning of the boat upstream the sediment transport gauging cross section,

-       drifting of the boat at flow velocity by turning off the motor engine. The hydrophone record is stopped after the boat crosses the sediment transport gauging cross section.

The sound of waves and the boat are reduced when drifting. Among all our hydrophone records (448), we noted that, only one survey (performed during high wind conditions) was characterized by waves oriented upstream and increasing the ambient noise. However, these noises do not affect acoustic power measurements because they produce noise in the low frequency band (<2 kHz) and we integrated bedload noise in higher frequency band (15-350 kHz). We also checked

if the lower limit for integration of the PSD affects the overall acoustic power recorded (Artus and Bouchard, 2021). The results of this work is clear: we miss a very small part of the bedload signal by integrating at 15 kHz. Among all the measurements, only one sample is greater than 1% of the reference acoustic (Figure 2). As expected, this sample appears when we decrease the integration lower limit to 1 kHz and it correspond to the windy weather conditions that we mentioned above.

[Figure]

Figure 4: Correlation between acoustic power integrated above 15 kHz and above varying lower limits (1, 2, 5 and 10 kHz).

b)    "What was the ambient noise level at the presence of the bars? How was the acoustic power correlated with these morphological changes?"

Immediately downstream of bars, bedload sediment transport is mostly equal to 0. In consequence, the noise of bedload particle impact should be very small or null. We illustrated this phenomenon with figure 10a (L. 339). In this figure, acoustic power is converted into bedload transport rate. The figure shows that an acoustic power of 1.1014 µPa² characterize the crest of the bar while it is equal to1.1012 µPa² downstream of the bar front. The acoustic power is not null in this area because the hydrophone records noise from the stoss side of the bar (located upstream) where sediment transport is active. So, the hydrophone measured a decrease of acoustic power downstream the bar fronts but this decrease of noise is under-estimated because it integers some noise coming from upstream. This phenomenon is accentuated by the low noise level downstream the bar.

c)    "Was the signal sensitive to the dune's height change?"

We saw that the signal is sensitive to change in sediment transport induced by location on the dune (figure 10b): the signal increases towards dune crest, but we don't have enough data to

know if the hydrophone is sensitive to the dune's height change. Dune morphology changes with hydraulic conditions so we need data from different hydraulic conditions to explore this question. Actually, work is in progress on this specific point in our team.

d)    "What was the bias of this signal towards the presence of gravel particles?"

Coarse particles produces noise at lower frequencies than fine particles. As a consequence, the Power Spectral Density of a coarser sediment mixture differs from a finer one. The median grain size is negatively correlated with central frequency of the spectrum (Thorne, 1986). As proposed by another referee, we decided to add a figure in the new version of the paper to illustrate this dependency.

**Specific comments (SC)**

*SC1*

*"L21-L23: Please correct the English. Something like: Although the parameters that control the self-generated noise still require analysis…"*

*"Although further work is needed to identify the parameters controlling sediment self-generated noise …"*

*SC2*

*"L23-24: The sentence does not contribute in the abstract and it is not clear. Does it mean that they are not able to measure if bedform migration is not happening?"*

It means that aDcp and hydrophone are both able to measure variation of sediment transport associated with the presence of dunes. "*Moreover, aDcp and hydrophone measurement techniques are sufficiently accurate to continuously measure bedload variations associated with dune migration.*"

*SC3*

*"L 48: Are you talking about multi-beam?"*

Here we refer to both single and multi-beam echo-soundings. We propose to add the following sentence: "*Moreover, these bathymetrical surveys are carried out simultaneously with sediment sampler measurements.*"

*SC4*

*"L 79: "adopted here as reference". Maybe delete and start new sentence giving information that these data are used as reference data because this method so far is the most reliable… or something similar."*

Agreed. We propose this sentence: "*Direct measurements of bedload sediment transport rates were performed using isokinetic samplers. This conventional approach was used to evaluate three indirect acoustic methods:*"

*SC5*

*"L92-L100: See general comment 2. All sub comments."*

Please see our response to these comments.

*"L102: You say two ADCPs. On Fig 2 I see only one and, in the description, I understood that only one was available per campaign."*

*"L102 "positioned" could be replaced with installed."*

You are right, there is a mistake in the previous version of the manuscript. There was only one aDcp mounted on the boat but aDcp changed from a campaign to another (Sontek, RDI).

"*Simultaneously with the BTMA measurements, an aDcp was installed on the boat (Fig. 2b). Measurements were performed using a Sontek Riversurveyor M9 (bi-frequency, 1 and 3 MHz) or a Teledyne RD Instruments Rio Grande (1.2 MHz).*"

*"L103: Which mode (IC-incoherent or SmartPulse HD?) was used in each case. Please add in the appendices (see, general comment 3 d.)."*

RDI aDcp uses coherent pulse system (Broadband) whereas Sontek aDcp use SmartPulse HD system that switches between Broadband and Narrowband system to provide the best data resolution. Most of the time, we used the Sontek device in manual mode fixing the pulse system to incoherent mode. We added the information in appendices B (P.21).

*"L124-L130: These are both kinematic models and essentially both describe a unit volume with its specific density moving with unit velocity. The first one is an assumption that the ADCPs are more sensitive to the particles and particle numbers and that by assuming characteristic particle Va would represent average in all 4 acoustic footprints. Considering g.c. 3c it is rather strange that you get better matching for Eq4 in Fig3b. But then the assumptions for Eq. 5 and the filtering are not correctly applied."*

We applied this equation based on a recent literature review (Holmes, 2010; Latosinski et al., 2017; Villard et al., 2005). As a starting point, we chose to fix the bedload concentration to the maximum bedload concentration according to van Rijn (1984) to compare to the computations performed for BTMA and DTM bedload rates. The thickness of the active layer was defined by van Rijn's (1984) equation as the saltation height. There was a mistake in the previous version of the manuscript on equation 6 ($d_s$=0.3 $D_*^{0.7}$ $T^{0.5}$ $D_{50}$). Fortunately this mistake was only in the text and not in the dataset. According to yours seminal comments (GC3j, SC19, SC20, SC33, SC35) we decided to analyze the effect of the apparent bedload velocity projection on kinematic models (Eq. 4 and Eq. 5). As expected, the concentration of bedload layer is probably overestimated when it is fixed to maximum bedload concentration ($c_0 = 0.65$), so, following your suggestion, we computed bedload concentration according van Rijn (1984):

$$c_b = 0.18\frac{T}{D_*} \; c_0;$$

Computed bedload layer concentration varies between 0.005 and 0.1 (0.03 in average). Bedload layer thickness ($d_s$) was computed with Eq. (6) and varies between $1D_{50}$ and $7D_{50}$ ($5D_{50}$ in average). In consequence Eq. (5) underestimates bedload rates. If we consider that van Rijn equations allow a good estimation of $c_b$ and $d_s$, $V_a$ seems to be underestimated and bedload rates computed using Eq. (5) underestimate BTMA bedload rates (24% of the dataset in the discrepancy ratio). By considering apparent bedload velocity without projection over flow direction, the kinematic model is able to better estimate BTMA bedload rates (41% of the dataset in the discrepancy ratio). We already seen in the paper that projected $V_a$ allowed a good estimation of bedload rates using Eq. (4). By using raw apparent bedload velocity, only 33% of the dataset are in the discrepancy ratio against 54% with projected $V_a$. So, as explained in the comment GC3b, the projection of apparent bedload velocity on flow direction allowed to filter incoherent values but it reduced the velocity magnitude. Consequently, Eq. (5) underestimates bedload rates and the use of raw apparent bedload velocity with a higher magnitude allowed to better estimate BTMA transport rates. Although, the application domain of Eq. (4) does not correspond to the characteristics of the Loire River, the decrease of projected $V_a$ seems to compensate the overestimation of bedload rates when the raw apparent bedload velocity is used.

[Figure]

*Figure 5: sensibility of $V_a$ post-processing to estimate BTMA bedload rates with both kinematic models employed in the paper, a) Equation 4; b) Equation 5.*

These results bring us to consider these two options of apparent bedload calculation in both kinematic model. Therefore, we could discuss the availability of our data processing and its consequence for using in kinematic models. The calibration between BTMA and aDcp with raw apparent bedload velocity is not possible and data need a filtering procedure as you proposed in GC3a. We propose to add:

- Equation of bedload concentration with appropriate modification in the Materials and Methods section (paragraph 3.2).
- Figures5a and 5b in the Results section 4.1. with the description of these results. "*Computed bedload layer concentration (Eq. 7) varies between 0.005 and 0.1 (0.03 in average). Bedload layer thickness ($d_s$) (Eq. 6) ranges between $1D_{50}$ and $7D_{50}$ ($5D_{50}$ in average). Bedload rates computed using Eq. (5) underestimate BTMA bedload rates with only 24% of the dataset in the discrepancy ratio (Figure 4b). By considering apparent bedload velocity without projection over flow direction, the kinematic model (Eq. 5) better estimates BTMA bedload rates with 41% of the dataset in the discrepancy ratio. Conversely, using raw apparent bedload velocity in the Equation 4, leads to only 33% of*

*the dataset in the discrepancy ratio against 54% with projected $V_a$. According to these results, the Equation 4 better describes the sampler bedload rates with projected apparent bedload velocity whereas raw apparent bedload velocity are preferred with the Equation 5."*

- The discussion of these new results in the paragraph 5.1. "*The results shown in Fig. 4a suggest that the equation 4 estimates sampler bedload rates if the projected bedload velocity is used. This kinematic model doesn't take into account the thickness or the concentration of the bedload layer and assumes that bedload transport never exceeds the size of a single particle assessed as uniform in terms of grain size (Rennie et al., 2002). These assumptions seems not appropriated for a sandy-gravel bed river. The active layer thickness should increase as suspended bed material load increase. Nevertheless, results are in agreement with BTMA bedload rates (Figure 4a). This can be explained by an underestimation of the apparent bedload velocity when it is projected along flow direction. On the other hand, Van Rijn (1984) defined the bedload layer thickness equal to the saltation height. The computed values of bedload layer thickness are coherent with other estimations made on comparable rivers (Conevski et al., 2020a). The equation 5 better estimates sampler bedload rates using the raw bedload velocity (Figure 4b). If we consider that $c_b$ and $d_s$ are well estimated by van Rijn equations, these results confirm that the projection of the apparent bedload velocity decreases the bedload velocity magnitude when the bedload direction differs from flow direction (e.g. bed slope effects). The influence of bedload velocity projection appear to be important when bedload are computed using kinematic models. Nevertheless, the calibration curve established seems to be in line with other studies. Although, the application domain of equation 4 does not correspond to the conditions of the Loire River, the decrease of projected $V_a$ seems to compensate the overestimation of bedload rates when the raw apparent bedload velocity is used. This is the opposite for Eq. 5 that takes into account bedload layer thickness and concentration. In this case, the projection of $V_a$ leads to underestimate bedload rates. Further works need to be done to improve the post-processing of $V_a$ by recently published filtering procedures (Conevski et al., 2019 and 2020a) and to estimate its effect on calibration curve and kinematic models.*"

**SC9**

*"L143-L150: Did you get better results in these surveys, when ADCP was close to the bed?"*

In this configuration, we did not compare aDcp measurements with BTMA sampled volumes. This protocol was used to investigate the capability of aDcp to describe bedload variation over dunes. We did not compare both configuration of the aDcp. It could be interesting to do this work by measuring with immerged aDcp and a floating aDcp to see if the protocol allow a better description of bedload fluxes by reducing the footprint. We supposed that this is the case, especially with the small dunes of the Loire River (footprint vs. dune size).

**SC10**

*"L150: Large window like that would do simple de-spike low pass filter. Not the noise but the outliers."*

That is correct. We changed "*noise*" by "*outliers*".

*SC11*

*"L151: All negative values refer to the average values, if they were negative you discard from the dataset? Or you applied some kind of filtering on the raw data as given in g.c. 3a? If so please cite Conevski et al 2019."*

We removed negatives values from the dataset after the data processing. We did not use any filtering on the raw data but, as mentioned above, it could be very interesting to test it in further works. We added your reference in the conclusion.

*SC12*

*"L170: So why did you adopt 0.5?"*

We mentioned line 182 that the discharge coefficient is equal to 0.5 for a perfect triangular dune shape. We tested two options according to our literature review (0.33 and 0.57).

*SC13*

*"L171: The accuracy of the echo sounding shall be very high. The problem is in the post-*

The accuracy of the echosounder is good but when you combine the accuracy of the DGPS, the echosounder and the navigation, the overall accuracy of the echosounder decreases. If you consider the RTK GPS accuracy and beam overture (6°) for the echosounder, it could be several centimeters. Based on our experience of the Loire River and on the papers above-mentioned, we chose 10 cm as a minimum dune height. It is nearly impossible to follow lower dunes especially during the celerity calculation post-processing. We modified the sentence by: "*Considering the accuracy of the bathymetrical echosounding relative to the dune size, the sinuosity of dune crests, and the representativeness of dune celerity, only profiles with a mean dune height of 0.1 m and more than 10 dunes are considered*".

*SC14*

*"L174: What are the dimensions of the hydrophone?"*

The hydrophone is 27 cm long and have a diameter of 3.8 cm.

*SC15*

*"L186: I assume the sampling was not synchronized?"*

Hydrophone drifts and BTMA sampling are not synchronized. We added this information to the paragraph 3.4.

**SC16**

*"L196: Is it the mean of all STDs?"*

Yes, "*mean value of 33 g.s$^{-1}$.m$^{-1}$*" is the mean value of all standard deviation.

**SC17**

*"L198: Again. Which of the two BTMA samplers were used in the average value?"*

Both BTMA samplers were used.

**SC18**

*"L203-204: I am not sure if this comparison makes sense. To me, it looks that your RMA reg model could fairly represent the entire dataset (Rennie + yours, in the range of R2~0.5). Would be interesting to see. Check g.c. 3e."*

By referring to your comment GC3e we agree that the variation of the apparent bedload is not only due to sediment grain size. We added a more general comment to this figure: "*The RMA regression presented here describes fairly well the dataset already published on several world large rivers*".

But, Rennie et al. (2017) showed this trend of increasing bedload rate with increasing sediment size (decreasing percentage of sand in the mixture).

**SC19**

*"L213: Why did not you calculate for each point separately? What does constant means? For all data? If this is the case then it makes sense eq 4 to perform better."*

We discarded this sentence because actually, as you mentioned in a specific comment above (*SC8),* the first kinematic model doesn't take into account the bedload thickness and concentration so it is not true to consider these parameters as constant. In the second model (Eq. 5), we initially considered bedload concentration as a constant (0.65) and bedload thickness as a variable (Eq. 6). So, only this last parameter was computed for each point. Please see also SC8 for related response.

**SC20**

*"L219-220: Could you clarify this sentence. Do you mean Eq5 is better than Eq4 or the opposite? Eq.4 does not assume any active layer thickness, but average particle size."*

We mean that, according to our data, Eq. 4 better estimates bedload rates obtained using BTMA than Eq. 5. Rennie et al. (2002) mentioned that this model assumes that the bedload transport

never exceeds the size of one particle. We thought that considering the active layer thickness proportional to the median sediment grain size reflected this assumption.

$$q_s ADCP = \frac{4}{3} \frac{D_{50}}{2} \rho_s V_{a\,proj};$$ (4)

$$q_s ADCP = 0.66\, D_{50}\, \rho_s V_{a\,proj};$$ (4)

And by comparing with Eq. (5) 0.66 could be assimilated to $c_b$ and $D_{50}$ to $d_s$

$$q_s ADCP = c_b\, d_s \rho_s V_{a\,proj};$$ (5)

That is why we made the assumption of bedload layer thickness proportional to median grain size in the Eq. (4).

**SC21**

*"Do you have something to say about the points down in the middle of the Fig3b?"*

These points are related to very weak transport conditions reached at low flows (Q<300 m$^3$.s$^{-1}$). It could be due to an underestimation of bedload with samplers as discussed in the general comment GC2e. We added this sentence to the paragraph 4.1: "*Some outlier data are observed for BTMA bedload discharge lower than 0.1 g.s$^{-1}$.m$^{-1}$. These points correspond to low flow conditions for which bedload samplers could under-estimate bedload fluxes (gap between the sampler mouth and the riverbed more significant)*".

**SC22**

*"L221-230: But that is reasonable. See g.c. 5a-b"*

OK. See GC5a and b.

**SC23**

*"L243: Does it involve the bedload concentration or just the velocity?"*

The transport stage parameter is calculated using equations mentioned in 3.2. Equations 7, 8, 9, 10 and 11 do not involve bedload concentration but mean flow velocity, shear velocity, grain size and sediment density ratio.

**SC24**

*"L254: The comparison is not consistent could you please clarify. The apparent velocity measures the velocity top layer or dynamical active layer, whereas the dune celerity is the mobility of the exchange event active layer, according to Church (2019)."*

We modified this paragraph accordingly to comments addressed by other referees. We added your comment that makes the sentence clearer. "*The apparent velocity measures the velocity top*

*layer or dynamical active layer (sediment transported over dune), whereas the dune celerity is the mobility of the exchange event active layer, according to Church and Haschenburger (2017)*".

**SC25**

*"L290: It is impressive if the DTM method somehow worked out in this morphology."*

Here we explained that it did not work in this specific case because of the presence of bars and the difference of integration transport scales between the methods. But, even in the presence of bars, dunes are well-developed.

**SC26**

*"Fig8. Was this data used in the calibration of the Eq14 and 15 as well?"*

Only data with a corresponding BTMA measurement were used in the calibration of equations 14 and 15. So, aDcp measurements of figure 8 are all used to calibrate equation 14, whereas only few points of the total hydrophone dataset were used to calibrate equation 15.

**SC27**

*"L303: bedload axis. Do you refer to a bedload active width?"*

Exactly. We changed that term according to your comment.

**SC28**

*"L313-L317 +Fig9: Did you use Eq14 to calculate the transport rate using the raw apparent velocities, estimating qs for every ensemble of va? If this is the case, the Eq14 is not calibrated over instantaneous values but over long averages, so It is not statistically sustained. Why don't you use only va? Fig10: same as the comment above. The equation is not calibrated for the instantaneous values of the instrument."*

We used Eq. 14 and Eq.15 to calculate transport rates from raw apparent bedload velocity and raw acoustic power respectively. Even if these equations were calibrated with long averages values, the goal here was to check if aDcp and hydrophone were able to record instantaneous bedload variations that would be consistent with bedform migration. We did not use raw values because we decided to keep the work done in the above paragraphs. We thought it would make the article more consistent. Do you think that calibration with instantaneous values is possible?

**SC29**

*"L325-328: Do you suggest that the hydrophones are sensitive to the bar formation?"*

No, we only suggest that the hydrophone is sensitive to the bedload variation induced by the front of the bar in a similar way to a dune crest (lee effect) but at a different scale.

**SC30**

*"L352: "bedload transport occurs over a very low thickness", please reformulate to: in weak bedload transport conditions the BTMA sampler most likely performed with reduced efficiency … and cite literature."*

"*Moreover, in weak bedload transport conditions, the BTMA sampler most likely performed with reduced efficiency initially calibrated to 50% (van Rijn and Gaweesh, 1992; Gaweesh and van Rijn, 1994; Banhold et al., 2016).*"

**SC31**

*"L378-384: This is very superficial information given in a way to suppress the ADCPs under the hydrophones. And it is only partially true. Please correct according to the general comments 3 and cite accordingly."*

Our idea was not to suppress the aDcp under the hydrophones. To our knowledge, to date there is no general equation available to compute bedload transport from apparent bedload velocity estimated on different rivers. We just wanted to highlight that, even if the results obtained from hydrophone calibration have to be taken with caution, it seems that a general tendency is emerging from Geay et al. (2020) and our results. Even if investigations were made in different rivers and using different hydrophones. Nevertheless we believe that the elements you gave in GC3 could seriously improve results given by the aDcp.

**SC32**

*"L386: After the filtering (Conevski, et al., 2019) negative values should not exist. But if there is beam heterogeneity this adcp value is rather incorrect and shell be eliminated not used as null. Also if somehow all the beams are sampling the dunes trough (lee side) the measurement is also incorrect if one aims to estimate the bedload transport over a cross-section."*

We agree that filtering could improve these results, especially instantaneous apparent bedload velocity. Here, we computed bedload with the calibration equation for all dataset even those which have been made without associated BTMA sampling. Where negative values were observed, we interpolated values from surrounding sampling points to compute the total bedload of the section. In consequence, we agree that our explanation cannot be use anymore.

**SC33**

*"L392-395: This is not true. See my other comments. Beside active layer with a height of D50 would assume only rolling of the particles."*

*"L395: You have observed only rolling of particles? Please send one video. I could only imagine this in incipient motion, although It is also impossible. On the contrary, (Lajeunesse, et al., 2010) and (Nino, et al., 1998) report that most of the particles are actually jumping, meaning at least 2D50."*

As mentioned above, Rennie et al. (2002) argued that this model assumes the transport never exceeds the size of a single particle. We referred to low flow conditions where we observed a kind of very thin bedload layer (about 1 particle in height). We are not able to measure the bedload thickness accurately from our videos in field condition. Nevertheless, we agree that considering the bedload thickness of a single particle size can be an underestimation of the actual thickness. This is why we try to discuss this model for high flow conditions.

**SC34**

*"L396: Please do not confuse suspension or suspended load with bedload and its active layer. Saltation is not a suspension."*

We agree that saltation (suspended bed material load) is part of the bedload. We understand that the text was unclear on this specific point and decided to modify according to your suggestion: "*The active layer thickness should increase with discharge that also increases suspended bed material load*".

**SC35**

*"L397-404: This should be reformulated because it is not true. Please see the general comments regarding the ADCP and the comment above. And cite (Conevski, et al., 2020) for the 3MHz"*

Please see GC8 for reformulation of this part of the discussion.

**SC36**

*"L408: Indeed, physical samplers sample the dynamical active layer, thus more comparable to the hydrophones and adcps. The usual approach is to integrate the sampled bedload rate over the active width and compare to the DTM."*

We agree. At line 448 we added: "*Moreover, physical samplers sample the dynamical active layer, thus more comparable to the hydrophones and aDcps*".

**SC37**

*"L421: This is very subjective. Conevski et al 2020 report 5min sampling per position using the M9. So in total 30 min, although no bars are presented in the study."*

This is the effective mean time needed to do the different protocols. For the aDcp method, we estimate this time for a survey without comparison with BTMA. During these surveys, aDcp was mounted on a small boat which is easier to handle and measurements were made on 6 sampling

points. The time mentioned for aDcp method take into account the time needed to position and anchor the boat at each sampling point (comment added). We based the sampling duration from literature available at the start of the study (Rennie et al., 2002; 25 minutes) and adjust to 10 minutes considering the variability of the sampling time in several studies summarized by Rennie et al. (2017). Same estimations were made for others methods.

*"L440: Could you elaborate on how this is not valid for the hydrophones if they are in a fixed position? In the same way, by use of the RTK one could move the ADCPs and estimate the BT velocity, or just subtract the boat velocity from the BT velocity measured by another device or the boat itself. Or if it is certain that one dune has passed below the ADCPs then using the filtering (Conevski, et al., 2019) the data shell be fairly accurate."*

The hydrophone was not used in a fixed position (please see our comments above). We made drifts of varying distance (depending of flow velocity) in order to reduce the ambient noise generated by water. These drifts integrate several dunes. As mentioned above we will use the filtering method proposed by Conevski et al. (2019) in our future works to improve these measurements (not possible during this study).

**SC39**

*"L452-456: Could you please reformulate. Do not really understand how the hydrophones are sensing the bars. I assume you mean change of the dunes which is just changing of the instantaneous transport and the shape of the dune… It could be another structure, not necessarily a bar… Is this correct?"*

Yes. We wanted to underline the role of the lee effect exerted by bars on dune development downstream of a bar front (Reesink et al., 2014). We simplified the text:

"*Moreover, as for the aDcp, the hydrophone also detects the theoretical pattern of bedload transport rates associated with bedform migration. As shown by Reesink et al. (2014), the lee effect generated by bar fronts influences the development of dunes downstream. Specifically, the hydrophone is able to record the decrease of the acoustic power immediately downstream the bar front and its progressive increase (traduced by the development of dunes at about 11h06, Fig. 10a). In the present study, dunes smaller than 0.4 m*"

**SC40**

*"L462: This is also partially true. The accuracy is within this range, but this stands for moving boat measurements. The laboratory data suggest lower values until 1mm/s check (Conevski, et al., 2020)."*

In Rennie et al. (2017) this accuracy is extracted from fixed field measurements. We acknowledge that in controlled conditions such as in laboratory studies, this precision can be lower. In our case, the scattering of the apparent bedload velocity for velocity lower than 1 cm.s$^{-1}$ (figure 3a) allowed

us to argue in this sense. But, filtering procedure that you mentioned could improve the accuracy by discarding points for this range of apparent velocity.

**SC41**

*"L469-L471: Please correct according to the general comments for the ADCP. If strongly correlated with the grain size, please show the correlation."*

The term strongly correlated is maybe too strong, but regarding figure 3a, for the same apparent bedload velocity, the bedload rate is higher when $D_{50}$ increase. We changed "*strongly correlated*" by "*dependent to*".

---

## Author Comment (AC3) · 8 Feb 2021

**General comments**

*"I think the conclusions regarding dune tracking are more specific to how the method was applied in this study and to this field site. Thus, I don't think that general conclusions about the accuracy or utility of that method can be claimed."*

You are right, DTM results are more specific to the study site especially to the dune scale that can vary with the presence of bars (Le Guern et al., 2019a). The application of the DTM was employed to have a second method (usually used in sandy-gravel bed rivers) to compare with aDcp and hydrophone measurements. Results would be improved if the study was centered around this method but to compare all these methods it is very time consuming in the field and protocols have to be adapted.

*"The authors state that acoustical power was not affected by water depth, but no data are presented to demonstrate that conclusion. Because acoustical power is expected to be affected by water depth. That should be demonstrated, rather than asserted."*

Please see Specific comment SC12.

**Specific comments (SC)**

_SC1_

_"L. 52-54: Poor sentence construction. Rewrite. Likely needs two sentences."_

"_In this work, we compare the efficiency of active and passive acoustic techniques to quantify bedload transport. The investigation took place in a reach of the Loire River (France) characterized by the presence of migrating bars and superimposed dunes (Le Guern et al., 2019b)._"

_SC2_

_"L. 57: What are "bedload axes"? Need to define or use more descriptive term."_

We agree that this term was unclear. We modified the sentence by: "_to estimate the accuracy of acoustic methods to measure cross sectional variations of bedload fluxes for various discharge conditions_".

_SC3_

_"L. 58-59: Not clear how this is different from #1"_

The third point aims at exploring bedload variations associated to bedforms with an aDcp and a hydrophone. Are they sufficiently accurate to record these bedload variations? The first point is related to the calibration of several acoustic methods with a direct measurement method without linking bedload to the presence of bedforms. We modified this sentence in order to be clearer: "_3) to investigate the capabilities of hydrophones and aDcps for capturing bedload variations along bedforms._"

_SC4_

_"L. 76: Why "theoretical" hydrophones?"_

This is theoretical position of hydrophone drifts because when we drift we were not able to follow exactly these theoretical lines. However, we agree that this term can be confusing. We decided to delete it.

_SC5_

_"L. 88: need full description of sampler (dimensions, weight, bag size, mesh size, etc.). Or reference to where that description can be found."_

"_Samplers consists of a sampling basket mounted on a frame. The sampling basket have a rectangular mouth of 0.05 m high and 0.085 m wide. Complete description of the sample can be found in de Vries (1979)._"

*"L. 94: This calibration factor seems to imply that the sampler is not actually "isokinetic". If it were truly isokinetic, it should not need a twofold correction factor."*

*"L. 97: This is an important reference for sampler calibration, but it's a "lecture note" that does not appear to be publicly available. If this reference is not available, these sampler details (calibration method and values) need to be described in the paper or an appendix."*

The reference is a textbook. We added the DOI in the reference list. Details of the calibration procedure are in two Dutch reports that we cannot find (Delft Hydraulics Laboratory, 1958 and 1969). But, additional elements are described in de Vries (1979). The first calibration tests consisted to compare the BTMA bedload catches with the average flume sediment transport. In a second time, tests were made by weighing catches in a sediment trap of the same size than sampler mouth. Sediment mixtures used during these tests was coarser than bedload of the Loire River with $D_{50}$ varying between 2.5 mm and 6.4 mm. Both test series concluded to the same calibration coefficient of 2 (efficiency of 50%). This means that actual transport is two times higher than those measured with the sampler. The average efficiency of a basket sampler as the BTMA sampler is about 45% (Hubbell, 1964). These tests established a calibration curve linking unit bedload transport rates with caught volumes of sediments. It is mentioned in Boiten (2003) that this calibration factor was not including the possible losses of sediments finer than 0.3 mm (size of the mesh). The sand loss was estimated during flume experiments with a similar samples by Banhold et al. (2016) to 50% in average but with a mesh size of 1.4 mm and varied sediment mixture ($D_{50}$=[0.8-10] mm). There is no detail about the hydraulic coefficient of the BTMA which is the ratio between velocity in the sampler and flow velocity in de Vries (1979). But, author argued that the BTMA construction promote the transport coefficient at the expense of hydraulic coefficient. In comparison, the Helley-Smith sampler tend to overestimate bedload with the ratio between the flume sediment transport and the sampler sediment transport that vary with flow velocity from 1.2 to 2.6 (Helley and Smith, 1971). In the new version of the manuscript we present the BTMA as a pressure differential sampler.

"*Suggested values of α and b were adopted from Boiten (2003) which mentioned that the trap efficiency factor not include the possible losses of sediment finer than 0.3 mm (mesh size opening). This calibration factor come from successive calibration procedures concluding to the same calibration coefficient of 2 (de Vries, 1979).*"

*"L. 106: Which approaches are considered the "empirical" and "calibrated"?*

*"I think everything described below is essentially empirical, but I'm not sure which approaches are calibrated?"*

The calibration approach is the approach which attempt to correlate bedload transport rates calculated from BTMA measurements with apparent bedload velocity measured by aDcp (Rennie et al., 2017).

*"L. 113-114: If the boat was static, wouldn't it be better to assume zero boat velocity than to assume the boat velocity equaled the bed (bedform) velocity?"*

The boat was almost static in the GPS referential (except some negligible lateral movements) but in the bottom track referential was mobile, due to mobile bed. When GPS data were missing, we considered the boat static in the GPS referential and we computed apparent bedload velocity directly from boat displacement in the bottom track referential.

*"L. 127-128: There is no ds in equation 4. Is there a typo in the equation, or in the sentence? It seems like there should be $d_s$ in the equation. If there is a constant ds, what is it?"*

You are right, this was not clear. According to other referee comment, we changed the part of the method by separating the 2 kinematic models and explaining that the first one assumes that maximum bedload thickness is a single particle:

*"$q_s ADCP = \frac{4}{3} \rho_s \, r \, V_{a\,proj} \times 10^3$;* (4)

*Where $r = D_{50}/2$ is the particle radius, $D_{50}$ is the median sediment diameter (m), $\rho_s$ is the sediment density (2650 kg.m$^{-3}$). In this model, it is assumed the maximum bedload thickness is a single particle.*

*$q_s ADCP = V_{a\,proj} \, d_s \, c_b \, \rho_s$;* (5)

*Where $c_b$ is the concentration of the active transport layer considered as the saltation height (van Rijn, 1984), and the van Rijn (1984) formulation was adopted to compute the active layer thickness ($d_s$) as a function of the hydraulic condition and sediment grain size:"*

*"L. 159-160: This is the only mention of interval time and it's not clear how many repeat profiles were measured to measure bedform celerity?"*

Bedform celerity was estimated from two repeated profiles for each sampling point. You can find mean interval time for each survey in the appendix D.

*"L. 187-188: As this is a new method, this should be explained in more detail. Especially because acoustic power is affected by transmission losses that are a function of distance. It may be that the losses are negligible in the range considered here, but that should be demonstrated rather than just asserted."*

Several tests were carried out to ensure that these acoustic power variations are not due to the variation of the distance between the hydrophone and the riverbed. A total of 20 drifts were performed with two hydrophones on the same horizontal location. The vertical location was

constant for hydrophone P1 (0.4 m depth) and variable for hydrophone P2 (from 0.4 m to 1.6 m depth). 3 drifts were done for each P1 and P2 configurations (except for P1 and P2 at the same depth, 2 drifts). When P1 and P2 are at the same depth, the acoustic power ratio is about 1.06, so the two hydrophones measured approximately the same acoustic power. This ratio doesn't evolve linearly with the proximity of river bed (depth ratio increasing). Even if the largest acoustic power ratio is observed for the largest depth ratio, it varies between 0.98 and 1.15 with a mean standard deviation of each configuration of 0.03 (see figure 1). This means that acoustic power can be considered as independent of the listening depth until 85% of the water depth.

[Figure]

Figure 1: Variable vertical position of hydrophone in the water column and associated acoustic power variation (red dots are the mean values of acoustic power).

**SC13**

*"L. 238: This should be in terms of qs as the dependent variable."*

We presented the equation with the acoustic power as the dependent variable to make the comparison with Geay et al. (2020) easier. This did not influence the result of the regression because we applied a Reduced Major Axis (RMA) regression. Effectively, a classical regression aim to examining the dependent variable (Y axis) and the deviation of Y values from the fitted line are minimized. The RMA minimizes the deviations of the observations from the fitted line in both X and Y directions (Davis, 2003).

**SC14**

*"L.307-308: While I think it's valid to use the hydrophone to look at the spatial distribution as done above, I don't think the direct comparison with the BTMA measurements is useful, because the hydrophone is calibrated to the BTMA. Some will "overestimate" and "underestimate" by definition of the calibration."*

Here, we considered that our calibration equation was verified and we compared results of this equation with BTMA measurements. Of course, the overestimation or underestimation depends of the calibration equation but we believe that it is interesting to see when the calibration could differ from BTMA sampling.

*"L. 346-347: Recently, Leary and Buscombe (2020), Ashley et al. (2020) and probably others have argued that repeat surveys with multibeam sonar are suitable for accurate measurements."*

Here we wanted to highlight that the samplers were the only direct measurement of bedload in the field that are used as a reference measurement. We propose to reformulate this sentence: "*Despite their lack of accuracy and their low spatial representativeness, isokinetic samplers allow a direct measurement of bedload and represents the only reference measurement of bedload in the field.*"

We agree that the multibeam echosounders are helpful for the measurements mentioned here. The main problem is due to the geometry of the Loire at this site (wide and shadow) that makes the multibeam echosounding pretty difficult to perform. This was already done on the Loire (Claude et al., 2014; Wintenberger et al., 2015) but on study sites that were smaller.

*SC16*

*"L. 371-373: The idea of a "global" calibration" for bedload by acoustic power is intriguing, but I have not yet had a chance to read the referenced paper in detail (it was only published this year). Given the wide variation in conditions that I think most would expect to work against a global calibration, a bit more discussion about why this might be possible would be helpful."*

The question of the possible existence of a global calibration curve was, as you mentioned, addressed in Geay et al. (2020). It could be possible because the intensity of acoustic power is proportional to the number of particle impacts, the size of these particles and the impact velocity. Theoretically, if we consider two different sediment samples (one finer than the other) that produce the same acoustic power and the same bedload rate, the finer one produce more noise at high frequencies and less noise at low frequencies. This is the opposite for the coarser sample. The high frequency noise produced by the finer sample is compensated by the low frequency noise of the coarser sample. Moreover, for the same bedload rates, the fine sample is composed by more particles than the coarser but can produce the same energy as energy is related to the number of impact and the size of sediment. These assumptions do not take into account the problem of acoustic wave propagation (Geay et al., 2019).

*SC17*

*"L. 406: Not clear what exactly is meant by this? Instantaneous?"*

Here "*punctual*" means local measurement. In other words, the DTM measure bedload over a longitudinal distance with potentially different hydraulic and sediment transport conditions whereas BTMA measure bedload at a sampling point that reflect local conditions.

*SC18*

*"L. 408: This is highly dependent on how the method is executed. If dune tracking is done at higher temporal frequency, it can provide"*

Yes, by extending the temporal frequency of DTM method, the registered bedload variation could be at the same temporal scale for both methods. But, the special scale will always differ.

**References**

Banhold, K., Schüttrumpf, H., Hillebrand, G. and Frings, R.: Underestimation of sand loads during bed-load measurements- a laboratory examination, in: Proceedings of the international conference on Fluvial Hydraulics (River Flow 2016), 11-14 July 2016, Saint Louis, USA, 2406 pp., 2016.

Boiten, W.: Hydrometry, IHE Delft Lecture Note Series, A.A. Balkema Publishers, Netherland, 256 pp, https://doi.org/10.1201/9780203971093, 2003.

Claude, N., Rodrigues, S., Bustillo, V., Bréhéret, J. G., Tassi, P., and Jugé, P.: Interactions between flow structure and morphodynamic of bars in a channel expansion/contraction, Loire River, France, Water Resour. Res., 50, https://doi.org/10.1002/2013WR015182, 2014.

Davis, J., C.: Statistics data analysis in geology, 3$^{rd}$ ed., John Wiley & Sons, New York, 2003.

de Vries, M.: Information on the Arnhem Sampler (BTMA), Internal Report n°3-79, Delft University of Technology, Department of Civil Engineering, Fluid Mechanics Group,1979.

Geay, T., Michel, L., Zanker, S., and Rigby, J. R.: Acoustic wave propagation in rivers: an experimental study. Earth Surface Dynamics, 7 (2), 537–548, https://doi.org/10.5194/esurf-7-537-2019, 2019.

Geay, T., Zanker, S., Misset, C., and Recking, A.: Passive Acoustic Measurement of Bedload Transport: Toward a Global Calibration Curve?, J. Geophys. Res.: Earth Surf., 125 (8), https://doi.org/10.1029/2019JF005242, 2020.

Helley, E., J., and Smith, W.: Development and calibration of a pressure-difference bedload sampler, U.S. Geological Survey Open-File Report, 1971.

Hubbell, D., W.: Apparatus and Techniques for Measuring Bedload, U.S. Geological Survey Water-Supply Paper 1748, 1964.

Le Guern, J., Rodrigues, S., Tassi, P., Jugé, P., Handfus, T., Duperray, A., and Berrault, P.: Influence of migrating bars on dune geometry, in: Book of Abstracts of the 6th Marine and River Dune Dynamics conference, 1-3 April 2019, Bremen, Germany, 157-160, 2019a.

Le Guern, J., Rodrigues, S., Tassi, P., Jugé, P., Handfus, T., and Duperray, A.: Initiation, growth and interactions of bars in a sandy-gravel bed river, in: Book of Abstracts of the 11th Symposium on River, Costal and Estuarine Morphodynamics, 16-21 November 2019, Auckland, New-Zealand, 226 pp., 2019b.

Rennie, C. D., Vericat, D., Williams, R. D., Brasington, J., and Hicks, M.: Calibration of acoustic doppler current profiler apparent bedload velocity to bedload transport rate, in: Gravel-Bed Rivers: Processes and Disasters, Oxford, UK: Wiley Blackwell, 209–233, https://doi.org/10.1002/9781118971437.ch8, 2017.

Wintenberger, C., L., Rodrigues, S., Claude, N., Jugé, P., Bréhéret, J.-G., and Villar, M.: Dynamics of nonmigrating mid-channel bar and superimposed dunes in a sandy gravelly river (Loire River, France), Geomorphology, 248,185-204, https://doi.org/10.1016/j.geomorph.2015.07.032, 2015.

---

## Author Response (AR2)

**Editor comments (EC)**

*EC1*

*L41-43"Here you say that acoustic techniques have only been applied in coarse rivers, but then at the end of this section you say that you will apply them in river with a fine bed. Is this the first time that this has been done, or can you refer to previous acoustic work in fine bedded rivers?"*

As far as we know, passive acoustic technique have never been applied in large lowland sandy fluvial environments. The only work that refers to sand particles is from Geay et al. (2020). They investigated 14 rivers where the median grain size ($D_{50}$) of bedload samples varies between 0.9 and 62 mm. They also measured the surface GSD (Grain Size Distribution) from surrounding emerged bars and, for example, the river with the finer sampled $D_{50}$ (0.9 mm, the Romanche River) have a surface $D_{50}$ of 31 mm. These wide range of the $D_{50}$ is not characteristic of sandy-gravel bed rivers. Moreover, the slope of this river (0.13%) is very high in comparison with the Loire River (0.02%). The same authors have demonstrated that the riverbed slope is an important parameter to take into account in the sound wave propagation especially for sand particles.

Earlier, Thorne (1984, 1986) investigated the passive acoustic technique in marine environments but this work was focus on gravel movement in turbulent tidal currents (particles larger than 2mm). Some laboratory works have been carried out with finer particles (0.3 mm; Thorne, 1985 and Thorne, 1986) but never in natural systems.

*EC2*

*L.54-55 "it might be useful to give an indication of grain size here, (e.g. sand-bedded river), so that it is clear by the end of the intro that you are focusing on sand-bedded channels".*

Detail of grains size distribution are given in the study site paragraph (few lines under this sentence). We modified the sentence to introduce sediment characteristics as you prescribed : "*The investigation took place in a reach of the Loire River (France), which is characterized by a sandy gravel bed evolving through bars and superimposed dunes migration (Le Guern et al., 2019b).*"

Yes it refers to the median diameter $D_{50}$ mentioned in the previous sentence. We specified this by modifying "it" by "The $D_{50}$".

*EC4*

*L. 66 "Briefly introduce the sampling first, e.g. 'For this work we measured flow and bedload at positions along the channel (Fig 1). At these sampling points, hydraulic conditions...' "*

*"During this work we measured the grain size distribution and flow characteristics at different locations along a cross section (Fig. 1). The riverbed is composed of a mixture of siliceous sands and gravels with a median diameter ($D_{50}$) of 0.9 mm. The $D_{50}$ varies between 0.3 and 3.1 mm with a standard deviation of 0.4 mm. The $90^{th}$ percentile of the sediment grain size distribution ($D_{90}$) is variable with a median value of 3.3 mm varying from 0.5 to 15.7 mm.* Hydraulic conditions varied according to discharge between 0.5 and 5.4 m for the water depth, and between 0.2 and 1.4 m.s$^{-1}$ for the water velocity (median water depth and water velocity are 1.9 m and 0.9 m.s$^{-1}$, respectively).*"*

*EC5*

*L. 67 "are these medians across the 100 times range in discharge? Given the wide variation in discharge, it might be useful to give the range of flow depths as well"*

These medians were extracted from aDcp measurements performed during bedload sampling surveys done with BTMA. We added the range of water depth and water velocity variation related to this discharge variation. "Hydraulic conditions varied according to discharge between 0.5 and 5.4 m for the water depth, and between 0.2 and 1.4 m.s$^{-1}$ for the water velocity (median water depth and water velocity are 1.9 m and 0.9 m.s$^{-1}$, respectively)."

**EC6**

*The appendixes are very useful, but you need to add a reference to them at an appropriate location in the text*

Done

**EC7**

*L. 88 "It's not clear to me whether all types of measurements were collected on all measurement dates. Are the isokinetic samplers the same as the BTMA?*

*Explain all abbreviations in the caption, even if they are also in the main text"*

It is difficult to show all types of measurements performed at different dates because sometimes we have 2 types of measurements, sometimes 3 at the same time. So, in order to make the figure more readable, we decided to put only BTMA measurements on the hydrograph because this is the reference measurement which is used to compare with other methods. We specified in the figure caption that this figure is related to bedload sampling surveys only.

**EC8**

*L. 108 "Explain that the suspended sediment stopped you seeing the bed"*

*We added this sentence in order to be clearer: "The increase of the water depth limits the light at the bottom of the water column and the addition of a mounted light did not improve the visibility because of particles in suspension."*

**EC9**

*L. 116 "min for the case with bedforms"*

Rennie et al. (2002) did not mention that the sampling time determined by their study is related to the presence of bedforms. Contrarily, Conevski et al. (2019) prescribed a sampling time for the

case without bedforms because when bedforms were present, they selected only the signal from the stoss side of the dune (eliminating values from the trough). Therefore, they did not need to register several dunes in order to take into account the variability of bedload due to the presence of these bedforms in the mean signal.

**EC10**

*L.125-128 "I think that this sentence could be phrased more clearly - the point that all measurements were made with the boat at anchor is important."*

The sentence was modified: "Even if the boat was anchored, the GPS signal was used in the Eq. 2 to correct apparent bedload velocity from small lateral displacements observed. When the GPS signal was poor or missing, $V_{GPS}$ was considered as null and $V_a$ resulted only from the bottom track signal $V_{BT}$ (representing 15% of the dataset)."

**EC11**

*Fig. 3 "Would it be helpful to show this figure with a logged x axis? It's hard to see how well the line fits most of the red data."*

We presented the Fig. 3 in a semi-log version because: 1) it allows to plot negative values of the apparent bedload velocity, 2) we though that it better illustrates differences between models of other rivers, especially the link with the grain size (as in Rennie et al., 2017), and 3) it is probably easier to define the lower detection limit mentioned L. 527. You can find the log-log version of the figure below.

[Figure]

*EC12*

*Fig. 6a "is this the PSD of a single measurement, or an average of all the measurements?"*

In the case of the Drau River and the Isère River (Geay et al., 2017; Petrut et al., 2018), this is the PSD of a single measurement. In our case, this is the median PSD of 450 measurements. We specified this in the figure caption: "Fig. 6: (a), Comparison of PSD from 3 rivers with varying D50 (PSD of the Drau River and the Isère River are extracted from a single measurement, PSD of the Loire River is the median PSD from 450 measurements)."

*EC13*

*L. 291 "Which apparent velocity are you referring to - the apparent velocity calculated from equation 13? Or using another method?"*

We are referring to apparent bedload velocity estimated using an aDcp: "The apparent bedload velocity estimated by aDcp is the velocity of the top layer velocity or dynamical active layer (sediment being transported over a dune), whereas the dune celerity is the mobility of the exchange event active layer, according to Church and Haschenburger (2017)."

These DEM are obtained from the interpolation of 2 single beam bathymetrical surveys using the same material as DTM method. "Fig. 8: Digital Elevation Models (obtained using natural neighbours interpolation of single beam bathymetrical surveys) showing location of sampling points with respect to bar location during: (a), survey of the 17/05/2018 (Q=604 m3.s-1) and (b), survey of the 19/12/2019 (Q=2050 m3.s-1)."

*L. 391 "did you actually calibrate the reduction in efficiency, or is this an estimate?"*

We did not perform calibration of the sampler. The sampler was initially calibrated during laboratory tests (de Vries, 1979). The efficiency of the bedload sampler was set at 50%, this means that bedload estimated by measured volumes sampled by BTMA are multiplied by 2. We observed from our camera dataset that, especially for low bedload conditions, the efficiency of the sampler is reduced because the sampler mouth is not well-posed on the river bed.

*The fact that this is the first demonstration that hydrophones work in a sandy gravel-bedded river is important, and you could make more of this in the intro and abstract.*

You are right. This part of our study is the most important part and must be to be highlighted. We added the following sentences in the abstract: "This study is the first work which attempted to use a hydrophone to quantify bedload rates in a large sandy-gravel bed river." And in the introduction: "This study aims to develop the use of passive acoustic technique in large sandy-gravel bed rivers for quantifying bedload rates and bedforms dynamics."